# Perlin Noise for Exploration in Reinforcement Learning

## Abstract

Reinforcement Learning (RL) enables agents to solve tasks by autonomously acquiring policies by interacting with the environment receiving sparse or noisy feedback in the form of a reward. However, achieving successful optimization in RL requires efficient exploration, which remains a significant challenge, particularly in continuous action spaces. Existing exploration techniques often exhibit limited state-space reach and fail to overcome local optima, resulting in suboptimal policies. Additionally, these techniques can cause erratic movements, posing risks when applied to real-world robots. In this work, we introduce a novel exploration strategy leveraging Perlin Noise, a gradient noise function that generates smooth, continuous disturbances, thus enhancing the agent's performance by promoting structured exploration and fluid motions. We quantitatively demonstrate the benefits of our approach compared to state-of-the-art methods, showing that it outperforms both unstructured and structured techniques in thorough experimental evaluations.

## 1 Introduction

It is well known that exploration in Reinforcement Learning (RL) is essential to successfully train the agent (Jiang et al., 2023). The policy is updated based on reward feedback to generate actions that control the agent to high-reward state regions. Visiting unseen and novel states in a broad range of the state space during this optimization is therefore essential to overcoming sub-optimal policies and converging to a high-performing policy. At the same time, the agent should explore the state space smoothly to prevent damage to itself (Raffin et al., 2022).

In the discrete action space, exploration has been addressed by various strategies such as epsilon greedy exploration (Amini & Solemany, 2008), Boltzmann exploration (Derthick, 1984; Amini & Solemany, 2008), or upper confidence bounds (Mizukami et al., 2017). Similarly, there has been extensive research on exploration strategies for continuous action spaces in the RL community (Fortunato et al., 2018; Osband et al., 2016; Plappert et al., 2018). Commonly RL methods rely on simple exploration strategies such as factorized Gaussian noise where in each decision step the action emerges by sampling from a Gaussian distribution (Schulman et al., 2017; 2015; Haarnoja et al., 2018). While these simple exploration strategies ensure good local exploratory behavior (see Fig. 1(a)), they might lack exploring relevant state-action regions that are far away from the initialization and might result in poor performance (Schumacher et al., 2023; Raffin et al., 2022). Additionally, the resulting motions are usually shaky and might damage agents such as robots (Raffin et al., 2022). Researchers have therefore proposed different exploration strategies to mitigate the aforementioned issues.

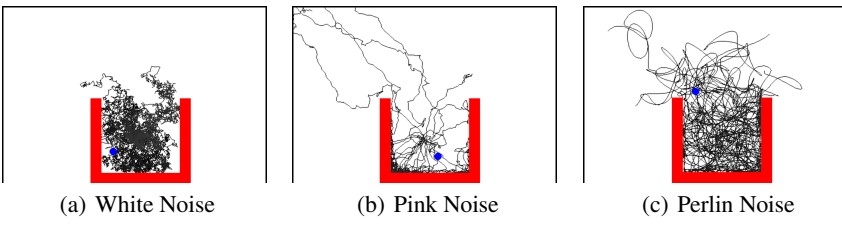

| (a) White Noise | (b) Pink Noise | (c) Perlin Noise |

Figure 1: An agent propelled by random actions sampled from different noises 'exploring' a 2D box.

Intrinsic motivation-based methods (Pathak et al., 2017; Burda et al., 2019) augment the objective function for the policy update with novelty functions that reward the agent if novel state spaces are visited. Alternative approaches apply exploration on the trajectory level by applying Gaussian noise in the parameter space that represents a trajectory rather than applying noise in each decision step (Otto et al., 2022; 2023; Celik et al., 2021; 2024; Li et al., 2023; 2024). However, these methods require additional treatment of the underlying optimization method by changing the objectives or introducing higher-level abstractions of the actions. In contrast, methods proposed by (Raffin et al., 2022; Eberhard et al., 2023; Hollenstein et al., 2024), propose simply exchanging the underlying exploration mechanism while maintaining the RL method.

This work proposes **Perlin noise for exploration in RL** for continuous action spaces, inspired by techniques from computer graphics (Perlin, 1985). Perlin noise generates smooth, temporally correlated disturbances by assigning random gradient vectors to points on a grid and interpolating between them, creating continuous transitions across space. The noise value at any given point is calculated by taking the dot product between the surrounding grid gradients and the vectors to that point, then blending these values using interpolation to ensure naturally flowing patterns.

We introduce a process to turn Perlin noise into a tractable distribution, allowing its usage as a drop-in replacement for White noise in Gaussian policies commonly used in RL algorithms such as PPO (Schulman et al., 2017), TRPO (Schulman et al., 2015), TRPL (Otto et al., 2021), and SAC (Haarnoja et al., 2019). This approach preserves the structure of these algorithms, requiring no modifications to their core objective functions or abstractions of actions.

In a qualitative analysis of Perlin noise compared to other exploration strategies, we show that it produces smoother and more coherent trajectories, leading to higher state-space coverage and broader exploration (see Fig. 1(c)).

We conduct extensive quantitative experiments on various benchmark environments, comparing Perlin noise to state-of-the-art exploration strategies, such as generalized State-Dependent Exploration (gSDE), White noise, and colored noise (Eberhard et al., 2023; Hollenstein et al., 2024), across multiple environments from diverse benchmark suites (Tunyasuvunakool et al., 2020; Yu et al., 2019; Kanagawa, 2023; Ellenberger, 2018; Towers et al., 2024). The results demonstrate that Perlin noise is capable of solving hard exploration problems, outperforming or performing on par with these baselines in both state-space coverage and task performance.

## 2 Background and Related Works

### 2.1 Exploration in Reinforcement Learning

For RL algorithms, it is important to effectively balance exploitation and exploration, which is crucial for discovering new behaviors in the environment to achieve optimal performance while avoiding local optima.

Action noise is the most common and simplest exploration method for continuous control, utilized by various RL algorithms (Schulman et al., 2015; 2017; Otto et al., 2022; Haarnoja et al., 2019; Abdolmaleki et al., 2018). Specifically, most agents leverage white noise, i.e., noise from an independent Gaussian distribution at each step, by sampling from a stochastic policy. While white noise can help in exploring new actions, it often leads to unstructured and jerky movements, which can be problematic in robotic applications (Peters et al., 2010; Otto et al., 2023). Maximum entropy RL, which typically also leverages white noise, further encourages exploration by adding an entropy term to the reward function, promoting policies that maximize both expected return and entropy. This approach results in more stochastic policies, thus enhancing exploration (Ziebart et al., 2010; Haarnoja et al., 2019).

Instead of white noise, Lillicrap et al. (2015) applies Ornstein-Uhlenbeck noise as actions noise. More recently, colored noises (Eberhard et al., 2023; Hollenstein et al., 2024) have also been introduced in deep RL, which incorporate temporally correlated disturbances instead of the traditionally used uncorrelated white noise. This temporal correlation leads to more structured exploration patterns, potentially enhancing exploration efficiency and performance in certain environments. With a similar goal, Rückstieß et al. (2008); Raffin et al. (2022) propose random sampling of a function for each episode that deterministically modifies action selection. Additionally, Schumacher et al. (2023)

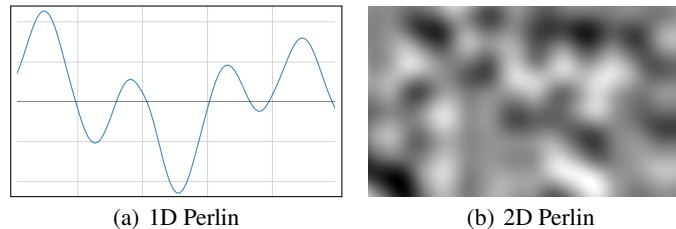

(a) 1D Perlin                         (b) 2D Perlin

Figure 2: Examples of 1D and 2D Perlin noise. (a) 1D Perlin noise shows smooth transitions along a single dimension, while (b) 2D Perlin noise creates a continuous, organic texture across two dimensions, often used for procedural texture generation.

explore learning correlations between action and state space dimensions. Alternatively, perturbing policy parameters instead of the actions themselves has been suggested (Plappert et al., 2018; Mania et al., 2018). A hybrid approach involves adding action noise to the entire trajectory by planning actions in the trajectory space (Otto et al., 2022; 2023; Celik et al., 2021; 2024; Li et al., 2023; 2024).

In addition to action noise, previous research (Thrun, 1992; Tang et al., 2017; Burda et al., 2018; 2019; Pathak et al., 2017) has incorporated novelty and intrinsic rewards to encourage exploration of previously unseen areas of the search space. Furthermore, insights from the bandit literature, such as Thompson sampling (Russo et al., 2018; Osband et al., 2016), can also be leveraged to enhance exploration strategies.

## 2.2 PERLIN NOISE

Perlin noise (Perlin, 1985), is a gradient noise function widely employed in computer graphics (Perlin, 1985; 2002; Bennett, 2019), simulations (Li et al., 2017), and scientific modeling (Ebert et al., 2002). It generates coherent, continuous, and seemingly random patterns that can be defined for spaces of arbitrary dimensions. Examples of Perlin noise spanned in one and two dimensions can be seen in Figure 2.

The process of computing Perlin noise at a point $\mathbf{x} = (x_1, \ldots, x_n)$ follows four steps:

**Identify Surrounding Lattice Points.** We identify the $2^n$ surrounding lattice points $\mathbf{i} \in \mathbb{Z}^n$ that form the corners of the hypercube containing $\mathbf{x}$. These points are given by the set

$$\mathbf{I} = \{\mathbf{i} = (i_1, i_2, \ldots, i_n) \mid i_k \in \{\lfloor x_k \rfloor, \lfloor x_k \rfloor + 1\} \text{ for } k = 1, 2, \ldots, n\}.$$

**Gradient Generation.** Perlin noise is spanned from randomly sampled gradients (see Figure 3(a)); for each lattice point $\mathbf{i} \in \mathbf{I}$, we uniformly sample a unit-length gradient vector $\mathbf{g_i}$ using a Pseudo Random Number Generator (PRNG), that is deterministic given a ($\mathbf{i}$, seed) pair via

$$\mathbf{g_i} = \text{PRNG}(\mathbf{i}, \text{seed}), \text{ with } ||\mathbf{g_i}|| = 1.$$

**Dot Products.** Each vertex gradient induces a hyperplane (see Figure 3(b)); for each lattice point $\mathbf{i} \in \mathbf{I}$, the dot product between the gradient vector $\mathbf{g_i}$ and the displacement between $\mathbf{x}$ and $\mathbf{i}$ is computed as

$$\mathbf{d_i} = \mathbf{g_i} \cdot (\mathbf{x} - \mathbf{i}) = \sum_{k=1}^{n} g_k(\mathbf{i})(x_k - i_k).$$

**Interpolation.** In order to have the final value smoothly vary across the edges of the different hyperplanes of the surrounding $2^n$ vertices, a smooth interpolation function is applied (see Figure 3(c)). We use smoothstep, as given by $\Phi(t) = 3t^2 - 2t^3$. The final Perlin noise value is then the weighted sum of the dot products

$$\text{Perlin}(\mathbf{x}) = \sum_{\mathbf{i} \in \mathbf{I}} \left( \mathbf{d_i} \prod_{k=1}^{n} \Phi(x_k - i_k) \right).$$

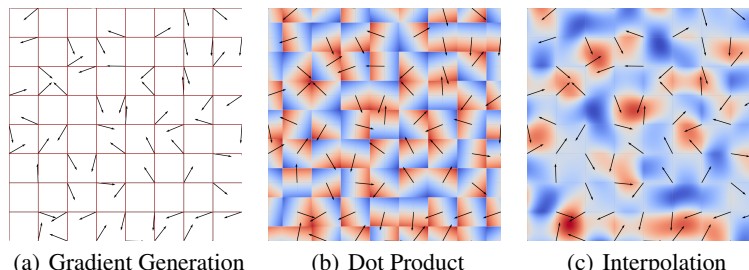

(a) Gradient Generation  (b) Dot Product  (c) Interpolation

Figure 3: Visualization of the steps involved in generating 2D Perlin noise. (a) Gradient vectors are randomly generated at the lattice points surrounding a given point. (b) Dot products are computed between each gradient vector and the displacement vector from the lattice point to the given point. (c) The final Perlin noise value is obtained by smoothly interpolating these dot products to ensure continuity across the grid.

We can regard any sampling from Perlin noise as sampling from a finite-dimensional subspace of an infinite-dimensional distribution, since

$$\text{Perlin}(x_1, \ldots, x_n) = \text{Perlin}(x_1, \ldots, x_n, 0, \ldots, 0).$$

This holds because, in the additional dimensions, $x_k = 0$ results in the interpolation $\Phi(0) = 1$, and the dot products vanish since $x_k - i_k = 0$ for $k > n$. Consequently, the noise function reduces to the $n$-dimensional case.

A pseudocode implementation for 2D Perlin noise can be found in Appendix G.

## 3 Perlin Noise for Exploration in Reinforcement Learning

Many RL methods, such as PPO (Schulman et al., 2017), TRPO (Schulman et al., 2015), TRPL (Otto et al., 2021), require calculating the likelihood of the current policy. This calculation becomes challenging when naively using Perlin noise, as the underlying probability density function generating these samples is unclear. Others generate gradients via reparameterization, e.g. SAC (Haarnoja et al., 2019). We can ensure correct operation and gradient generation for both these cases by ensuring the Perlin samples to follow the Gaussian policy distribution normally used in these methods. This also ensures Perlin-based exploration is usable as a drop-in replacement.

### 3.1 Sampling Actions using Perlin

Perlin noise on its own does not follow a Gaussian distribution and is not related to it. To use Perlin noise for smooth exploration, we wish to generate noise that aligns with the policy, which we continue to model as a Gaussian distribution. For a given state $s_t$, sampling from a Gaussian policy with mean $\mu_\pi$ and covariance $\Sigma = L_\Sigma^T L_\Sigma$ is formalized as

$$a_t = \mu_\pi + L_\Sigma \epsilon, \quad \epsilon \sim \mathcal{N}(0, 1),$$

where $L_\Sigma$ is the Cholesky factor of the covariance. Sampling $\epsilon \sim \mathcal{N}(0, 1)$ leads to generating samples from the parameterized Gaussian policy. However, for sampling actions from a policy that applies Perlin noise, we reformulate the sampling process to

$$a_t = \mu_\pi + L_\Sigma \epsilon, \quad \epsilon \sim \mathcal{P}(x, y), \quad \text{where } \mathcal{P}_i(x, y) = \text{Normalize}(\text{Perlin}_i(x, y)).$$

We sample from Perlin noise with $x = t \cdot k_{speed}$ and $y = i \cdot k_{offset}$, where $t$ is the timestep index and $i$ is the action dimension index. These define the sampling line, as depicted in Figure 4. Since Perlin passes through 0 at each vertex, we consider 2D Perlin noise, where the parameter $k_{offset}$ shifts the sampling line away from integer coordinates, and the parameter $k_{speed}$ defines how fast the noise changes over time.

While $k_{speed}$ is treated as a tunable hyperparameter, $k_{offset}$ is set to a constant value (e.g., $\pi$), as in our experiments it showed negligible impact on performance for reasonable choices.

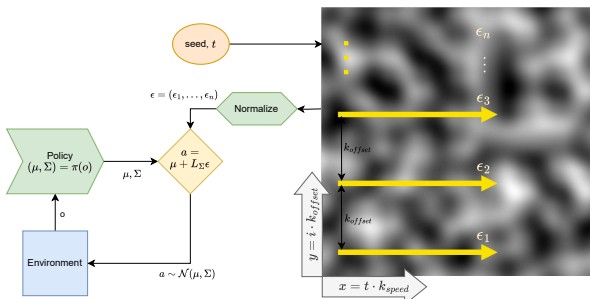

Figure 4: A schematic illustrating the integration of our Perlin noise sampling method into an existing RL algorithm (left), alongside a visualization of the sampling lines in 2D Perlin noise (right).

The computational complexity of our approach to sampling via Perlin noise is $O(a \cdot n \cdot 2^n)$, where $a$ is the action dimensionality and $n$ is the dimensionality of the spanned Perlin noise (fixed at 2 in our case). The added overhead is negligible compared to other noise generation methods.

The function *Normalize* transforms Perlin noise samples to match standard normal moments, as described in the next section. This enables its use in RL methods, supporting log-likelihood gradient estimation (e.g., PPO, TRPL) and reparameterization-based approaches (e.g., SAC) without modifying existing procedures.

### 3.2 THE NORMALIZATION FUNCTION FOR PERLIN NOISE

In order to establish the existence of an appropriate normalization function, we consider that Perlin noise inherently centers itself around zero, a characteristic stemming from its generation mechanism (derivation in Appendix D.1). Further, the autocorrelation function $\rho(k)$, governing the relationship between two samples $x_i$ and $x_j$, clearly exhibits a diminishing trend as the lag parameter $k = j - i$ approaches infinity. Consequently, the Central Limit Theorem (CLT) becomes applicable, asserting that the expected empirical mean of our samples converges to $\mu = 0$. Furthermore, due to Perlin's construction, it restricts its moments to finite orders beyond the second (derivation in Appendix D.2). Making use of Asymptotic Normality, we can deduce that the empirical variance will tend towards a constant as the sample size grows sufficiently large.

We construct Normalize($x$) as a polynomial expansion of degree $M$

$$\text{Normalize}(x) = \sum_{n=0}^{M} c_n x^n.$$

This parameterization is justified as Normalize($x$) is an analytic function (derivation in Appendix D.3). Since Perlin already has $\mu = 0$, it follows that $c_0 = 0$.

We find a suitable Normalize($x$) function via the optimization problem

$$\min_{c_1, c_2, \ldots, c_M} E(c_1, c_2, \ldots, c_M),$$

where $E$ is the error function that enforces the empirical moments of the transformed Perlin noise to match the theoretical moments (mean, variance, skewness, etc.) of a standard normal distribution $\mathcal{N}(0, 1)$. The error function is defined as

$$E(c_1, c_2, \ldots, c_M) = \sum_{m=1}^{k} \left( \frac{1}{N} \sum_{i=1}^{N} \text{Normalize}(x_i)^m - \mu_m \right)^2,$$

where $\mu_m$ are the theoretical moments of $\mathcal{N}(0, 1)$, and $x_i$ are the Perlin noise samples.

This optimization must only be performed once to find a suitable Normalize function and is not part of the training or inference loop. We experimentally validate this normalization and provide the results in subsection 3.3. Our implementation of this normalization function is available on GitHub[1]. We found an expansion to first order to be sufficient to achieve accurate normalization.

---

[1] https://github.com/perlin-rl/Perlin_RL

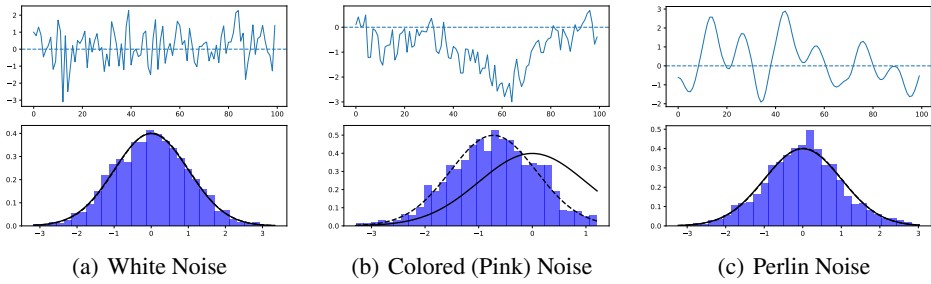

(a) White Noise          (b) Colored (Pink) Noise          (c) Perlin Noise

Figure 5: The top diagrams illustrate action trajectories for different noise types sampled from a static Gaussian policy $\mathcal{N}(0, 1)$. The bottom diagrams present histograms, with the solid black curves representing the true policy distribution and the dotted black lines a bell curve matched to the actual samples.

### 3.3 COMPARISON OF SAMPLING BEHAVIOR TO EXISTING NOISES

To analyze the sampling behavior of various exploration noises, we first focus on their application to a static Gaussian policy distribution $\mathcal{N}(0, 1)$, without considering an RL setup. The diagrams in Figure 5 show the resulting samples. The top diagrams depict 100 steps of the trajectory over time, while the bottom ones display histograms of 2000 sampled steps, with the solid black curves representing the true policy distribution. Additionally, the dotted black lines in the histograms match a bell curve to the actual samples, providing a visual comparison of how well the noise types conform to the expected distribution. We provide a Google Colab[2] that allows testing various parameters and replicating these results.

**White Noise** (Figure 5(a)) is the default noise used in Random Exploration (REX). As expected, the empirical mean and variance align closely with the parameters of the Gaussian policy. However, because the disturbances are sampled independently at each time step, the resulting motions are jerky and unstructured. In physical systems like robotics, these sudden, erratic movements can cause damage. Furthermore, White noise fails to achieve significant displacement in the state-space, limiting the range of exploration, as shown in Figure 1(a), where a particle driven by White noise spends most of its time in a limited area.

**Colored Noise** (Figure 5(b)), such as Pink noise, introduces temporal correlations into the disturbances, allowing for more structured movements and greater displacement in the state-space. While colored noise theoretically converges to the true policy distribution as sample size approaches infinity, in practice, with finite rollouts, there can be a significant mismatch between the empirical and true policy parameters. This can lead to suboptimal exploration, as the agent may over-explore or get stuck, as shown in Figure 1(b), where a particle driven by Pink noise spends much of its time against walls.

**gSDE** cannot be evaluated independently, as its noise generation depends on the latent activations of the policy network's last hidden layer. Consequently, the behavior of gSDE varies between tasks and across different stages of training. While gSDE provides smooth, structured motions, its realized variance depends on the neural network's architecture and weight initialization, making it less consistent. The periodic resampling mechanism introduces some discontinuity, but overall, gSDE smooths the exploration trajectory compared to White and Colored noise.

**Perlin Noise** (Figure 5(c)) provides a smooth and temporally correlated alternative to both White and Colored noise without relying on periodic resampling. The smoothness of the trajectory is controlled by a speed parameter, making it easier to tune and adapt across environments. Unlike gSDE, Perlin noise is not dependent on the architecture of the neural network or the latent activations, leading to more predictable and consistent behavior. As shown in Figure 5(c), Perlin noise achieves structured exploration with minimal drift from the true policy parameters, even in finite rollouts. This makes it particularly suited for on-policy exploration. In the exploration box scenario (Figure 1(c)), Perlin noise successfully escapes the box, similar to Pink noise, but without the drawbacks of getting stuck against walls.

---

[2]https://colab.research.google.com/drive/1-t7WmGCwEgkZWuriRN3dsuy5fmU34v9x

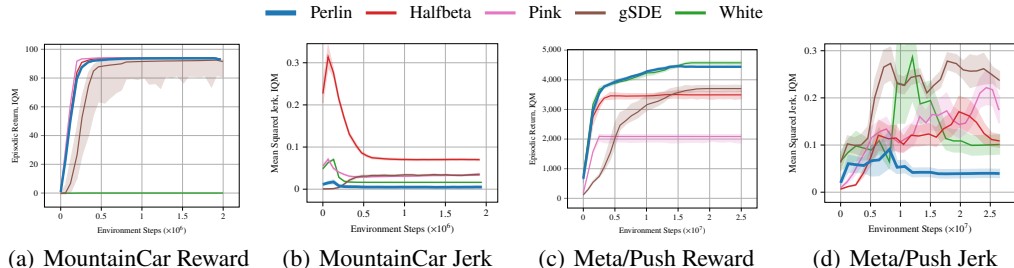

|(a) MountainCar Reward|(b) MountainCar Jerk|(c) Meta/Push Reward|(d) Meta/Push Jerk|

Figure 6: Achieved episodic reward and smoothness (measured as mean squared jerk, lower is better) on MountainCarContinuous-v0 and Metaworld/push-v2.

## 4 EXPERIMENTS

We evaluate the performance of Perlin noise-based exploration against existing methods, including generalized State-Dependent Exploration (gSDE), White noise, Pink noise, and HalfBeta noise. We use the term HalfBeta noise to refer to colored noise with $\beta = 0.5$, which was found to be the optimal coefficient for on-policy settings in Hollenstein et al. (2024). Benchmarking exploration capabilities can be particularly challenging, as many established environments were designed with current algorithms in mind. To address this, we introduce custom maze environments that present difficult exploration tasks. Additionally, we assess the performance of Perlin noise across a wide range of standard benchmark suites to ensure a comprehensive evaluation against state-of-the-art (SOTA) methods.

We use on-policy reinforcement learning, specifically Proximal Policy Optimization (PPO), for all experiments. The performance of the noise on each environment was evaluated using 20 runs (only 10 for MetaWorld), each with a different random seed. To calculate stratified bootstrapped confidence intervals, we use the methodology proposed by rliable (Agarwal et al., 2021). The resulting interquartile mean (IQM) and confidence intervals (CI) for all tested environments are presented in the appendix (see Appendix A). Our summary bar chart shows the mean and standard error (SE) across all evaluated environments in the specific suite. Here, we use the regular mean instead of the IQM, as we do not treat exceptionally good or poor performance across entire environments as statistical outliers. This contrasts with handling over- or under-performance in a single run within an environment.

In addition to reward performance, we measure the smoothness of the action trajectories, quantified by the mean squared jerk. Lower jerk values indicate smoother actions, which are desirable for many physical systems and tasks requiring stable, continuous actions. Similar to reward, smoothness is evaluated using IQM and stratified bootstrapped confidence intervals (CI). A formal description of Mean Squared Jerk can be found in Appendix B.1, the results in Appendix B.2.

To ensure fair evaluation and reduce the risk of overfitting hyperparameters (HPs) onto specific environments, we emphasize the importance of using shared hyperparameters across algorithms and environments wherever feasible. Overfitting HPs could lead to misleading comparisons, where methods may appear to perform better due to environment-specific tuning rather than the intrinsic quality of the exploration strategy. Therefore, we have made a deliberate effort to use shared HPs across all algorithms and environments as much as possible.

Our choice of HPs is based on prior work to ensure relevance and generalizability. Specifically, for PyBullet, we follow the hyperparameters used by Raffin et al. (2022). For MetaWorld, we adapted the settings from Li et al. (2024). For general environments, such as MujocoMaze and DMC, we rely on the findings from Hollenstein et al. (2024).

The episodic return achieved over time on every environment tested can be found in Appendix A, the smoothness of generated actions for all environments in Appendix B. For a complete overview of the hyperparameters used in our experiments, see Appendix C.

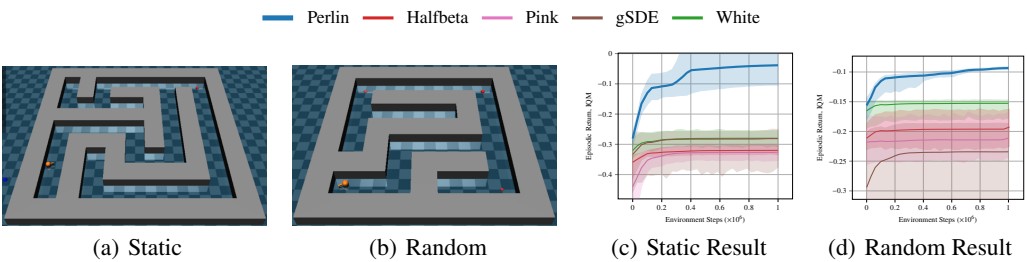

(a) Static          (b) Random        (c) Static Result     (d) Random Result

Figure 7: Static and random goal mazes used to benchmark Perlin noise against other exploration methods, showcasing the resulting performance for difficult exploration challenges.

### 4.1 MOUNTAINCAR (GYMNASIUM)

The Mountain Car Continuous environment (Moore, 1990) from Gymnasium (Towers et al., 2024) is a deterministic MDP where a car starts at the bottom of a sinusoidal valley. The goal is to accelerate the car strategically to reach the top of the right hill, which requires continuous and consistent actions. The challenge lies in overcoming the gravitational pull by building up momentum, making this task particularly difficult for exploration methods that rely on random, chaotic actions.

In our tests (see Figure 6(a)), White noise performed very poorly, as it failed to apply the consistent accelerations needed to push the car uphill. While environment-specific hyperparameter tuning could likely improve its performance, we conducted all tests without such adjustments to ensure consistency. On the other hand, all other methods, including Perlin noise, performed similarly well, showing stable results in this challenging task. Moreover, in Figure 6(b), we show the smoothness of generated action trajectories, measured by the mean squared jerk (lower values indicate smoother trajectories). Perlin noise outperforms all other methods, producing the smoothest trajectories overall. Notably, there were significant differences in performance between algorithms. For example, the HalfBeta method performed exceptionally poorly, producing extremely jerky actions.

### 4.2 CUSTOM MAZES

To demonstrate Perlin noise's effectiveness in difficult exploration tasks, we created two custom mazes based on MujocoMaze (see Figure 7 (a,b)). One maze features a static goal position, and the other has a goal that is randomly chosen from three possible locations. The agent always starts in the bottom left corner, with the goal in the top right for the static maze, or in any other corner for the random maze. We found (see Figure 7 (c,d)) that the exploration challenge in this environment is sufficiently hard, causing all baseline methods tested to fail in learning a reliable policy. In contrast, Perlin noise enabled the agent to successfully solve these environments. This shows that there are exploration challenges that require more advanced techniques than current SOTA methods, and Perlin noise can provide such a solution.

### 4.3 MUJOCOMAZE

We tested Perlin noise on several environments from the MujocoMaze suite (Kanagawa, 2023). These environments, which include agents such as simple dots, ants, and swimmers navigating various mazes, were designed with SOTA methods in mind, making dramatic improvements hard to achieve. While most tasks are relatively easy for modern RL algorithms, they remain useful for evaluating exploration methods. In Figure 8(a) we can see how Perlin noise performed consistently well across all tasks, maintaining stability and solving the environments without any performance degradation. Pink noise performed the worst, while the other methods showed similar results, with Perlin noise slightly outperforming them.

### 4.4 DMC

We tested Perlin noise on several environments from the DeepMind Control (DMC) Suite (Tunyasuvunakool et al., 2020). The DMC suite is a standard collection of physics-based simulation

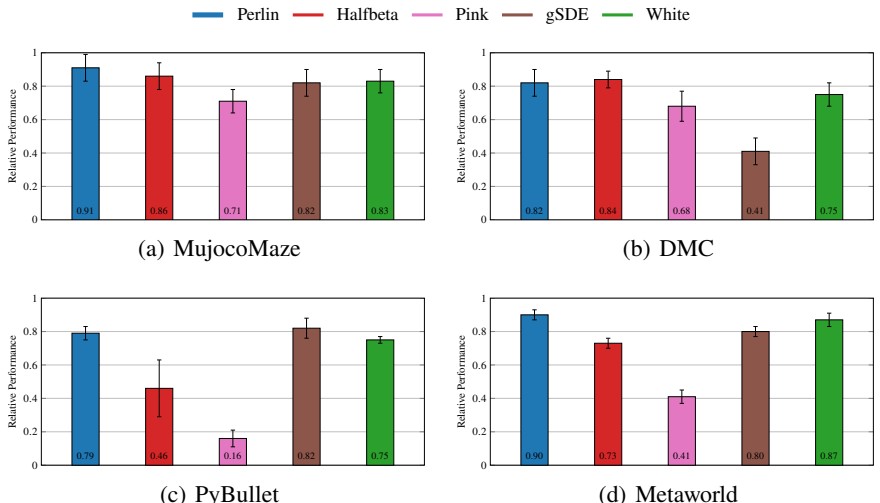

Figure 8: Aggregate results showing the mean and standard error (SE) of episodic reward across entire suites, with performance for each environment normalized relative to the best-performing algorithm.

environments powered by the MuJoCo engine, designed to test continuous control tasks. The chosen tasks are inspired by the evaluations performed in Hollenstein et al. (2024). In our experiments (see Figure 8(b)), Perlin noise performed well overall, coming second only to HalfBeta. Notably, Perlin noise outperformed gSDE, which demonstrated poor results across most DMC environments.

### 4.5 PyBullet

The PyBullet Gymperium (Ellenberger, 2018; Coumans & Bai, 2016) suite provides an open-source implementation of continuous control environments commonly used in reinforcement learning, originally based on the OpenAI Gym MuJoCo tasks. In our experiments (see Figure 8(c)), we tested Perlin noise on four environments, the same tasks used in the gSDE paper (Raffin et al., 2022) to evaluate on-policy performance. gSDE performed best on these tasks, followed closely by Perlin noise, significantly outperforming Pink noise and HalfBeta, both of which performed poorly in these tasks.

### 4.6 Metaworld

MetaWorld (Yu et al., 2019) is an open-source simulated benchmark designed to advance meta-reinforcement learning and multi-task learning, comprising 50 diverse robotic manipulation tasks. These tasks take place in a shared tabletop environment featuring a simulated Sawyer robotic arm interacting with various everyday objects. The benchmark is particularly well-suited for exploring generalization and meta-learning due to its structured setup and diverse task distribution.

In our experiments, we focused on training policies for individual tasks, evaluating performance on each task separately to assess the effectiveness of different exploration strategies. The results indicate that Perlin noise consistently outperformed other methods, achieving the best overall performance across the tasks. White noise followed closely as a second option, demonstrating solid performance but with less consistency compared to Perlin noise. In contrast, the other methods tested, tended to exhibit somewhat unreliable performance.

As an example, we present the smoothness of action trajectories (measured by mean squared jerk, where lower is better) for the Metaworld/push-v2 environment. In this case, both Perlin noise and White noise achieved similar rewards (see Figure 6(c)), with Perlin noise exhibiting the lowest jerk (see Figure 6(d)), thus producing the smoothest action sequences.

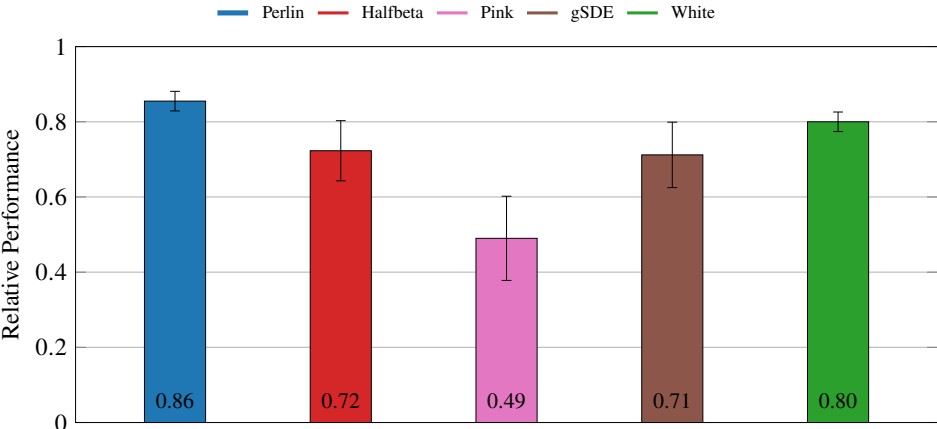

Figure 9: Aggregated results showing the mean and standard error (SE) of relative performance across all suites, excluding MountainCar and our custom mazes.

## 5 DISCUSSION

As demonstrated in Figure 9, Perlin noise consistently delivers strong and reliable results across all tested suites. While Perlin noise is not always the optimal choice, its consistent and generally favorable performance makes it a dependable exploration method for various applications without the need for extensive tuning or task-specific adjustments.

In subsection 3.3, we had showed that Perlin noise produces significantly smoother trajectories than other methods, and this is now validated in our experiments. As shown in Appendix B, Perlin noise consistently shows lower jerk compared to other methods. While Pink noise also exhibits low jerk, its smoothness often comes at the cost of poor task performance, as its learned policies sometimes tend to remain close to a null policy. Perlin noise, on the other hand, achieves both smooth action generation and high task performance, making it a well-balanced choice for structured exploration.

Additionally, in the earlier analysis, we demonstrated that Perlin noise has superior state-space coverage compared to White and colored noise. This is now reflected in its superior performance on the custom and suite-provided maze environments, where better state-space reach is crucial for effective exploration.

## 6 CONCLUSION & LIMITATIONS

Exploration remains a critical component in the success of reinforcement learning (RL) algorithms, as it drives agents to visit novel and high-reward states during training. In this work, we introduced a novel exploration strategy that utilizes Perlin noise, a smooth, temporally correlated gradient noise function. Perlin noise distinguishes itself by its ability to provide structured exploration. Unlike conventional noise strategies like White noise, which can result in jerky, erratic movements, Perlin noise promotes fluid motion, making it particularly suitable for tasks where smooth and stable actions are required, such as in real-world robotic applications. As demonstrated in our experiments, Perlin noise offers high state-space coverage, ensuring the agent explores effectively across a broad range of tasks. Our approach was validated across various benchmark suites, showing competitive performance when compared to state-of-the-art exploration strategies such as gSDE and colored noise.

Despite these advantages, Perlin noise has limitations. Its inherent smoothness, while beneficial in many tasks, may hinder performance in environments where abrupt, high-frequency actions are necessary for success.

## Reproducibility Statement

We provide detailed documentation of hyperparameters, shared across environments to avoid overfitting, in Appendix C. Our results are reported using stratified bootstrapped confidence intervals and interquartile means (IQM), ensuring statistical robustness.

The implementation of our novel Perlin noise-based exploration mechanism, compatible with Stable Baselines3 (SB3) (Raffin et al., 2021), is available in an open-source repository[3].

Additionally, a Google Colab notebook[4] allows interactive testing of noise parameters, replicating the sampling behavior shown in subsection 3.3.

---

[3]https://github.com/perlin-rl/Perlin_RL
[4]https://colab.research.google.com/drive/1-t7WmGCwEgkZWuriRN3dsuy5fmU34v9x

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

# A RESULTS

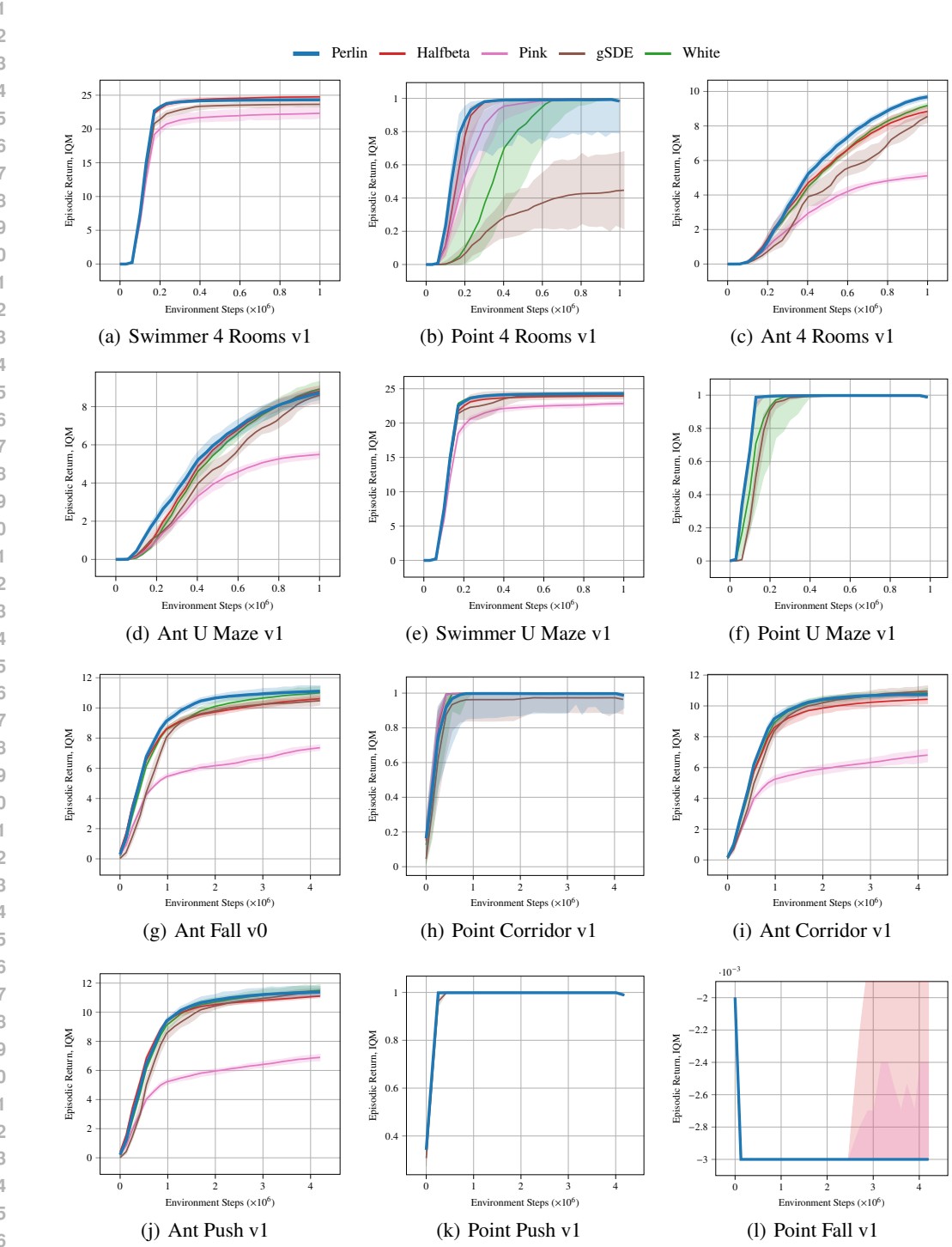

Figure 10: Results on selected environments from MujocoMaze (Page 1).

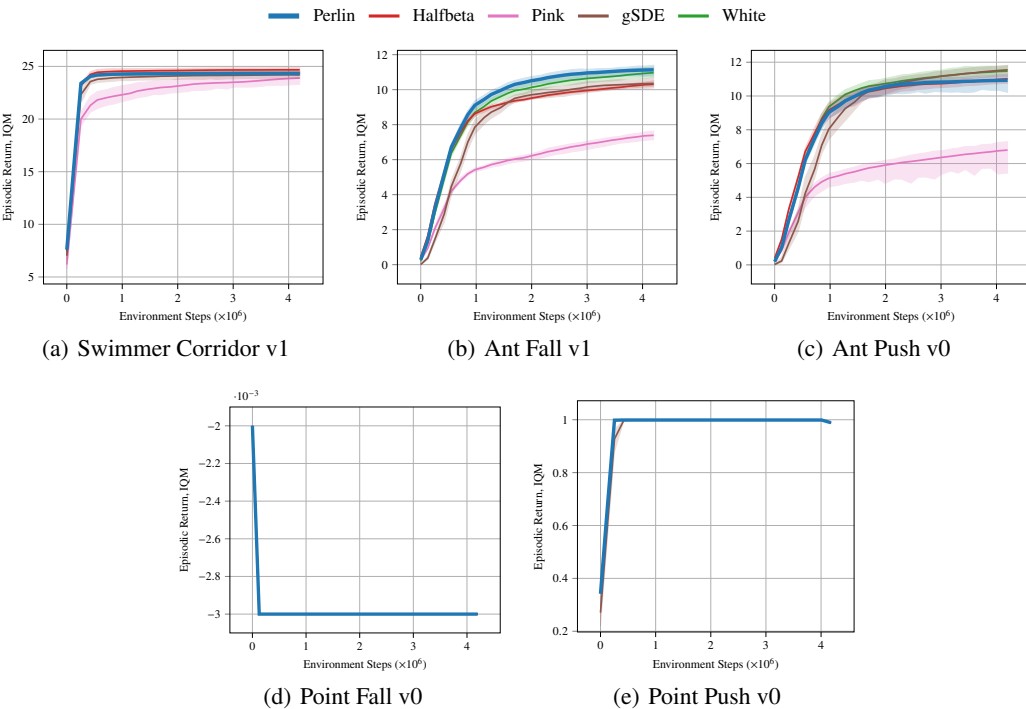

Figure 11: Results on selected environments from MujocoMaze (Page 2).

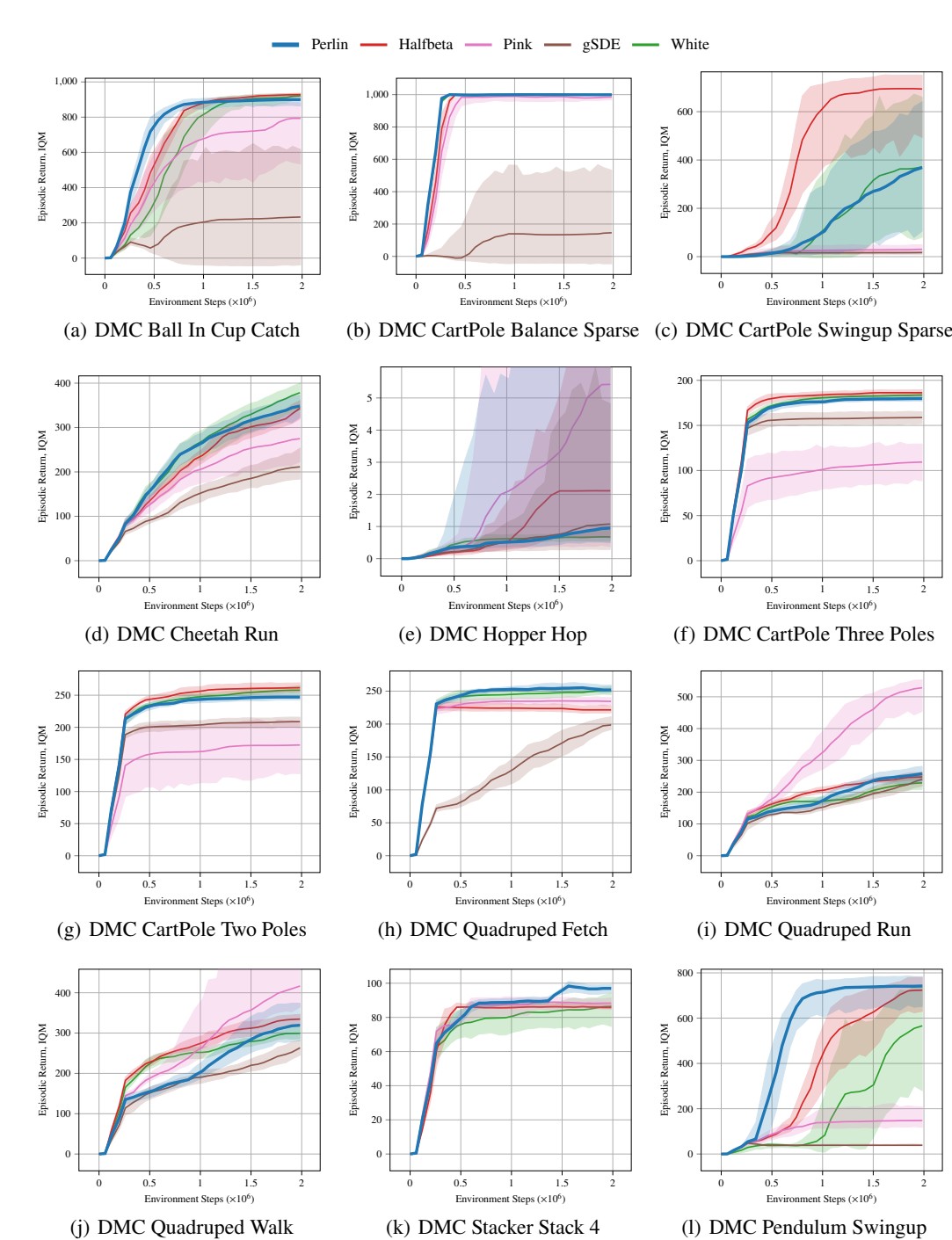

Figure 12: Results on selected environments from DMC (Page 1).

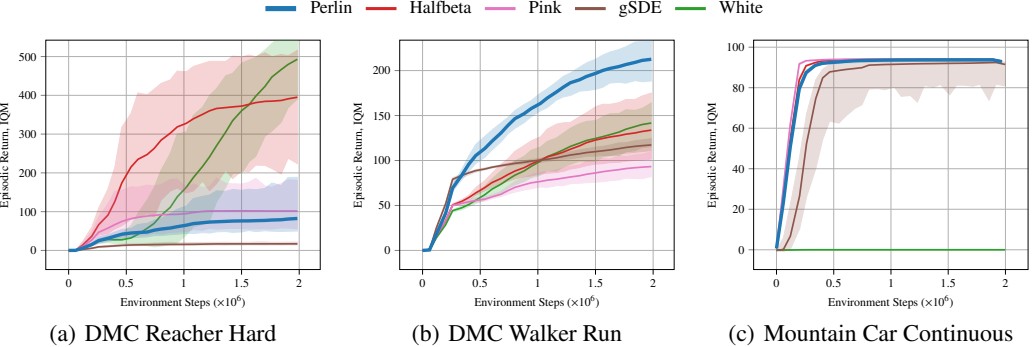

(a) DMC Reacher Hard  (b) DMC Walker Run  (c) Mountain Car Continuous

Figure 13: Results on selected environments from DMC (Page 2) and MountainCarContinuous.

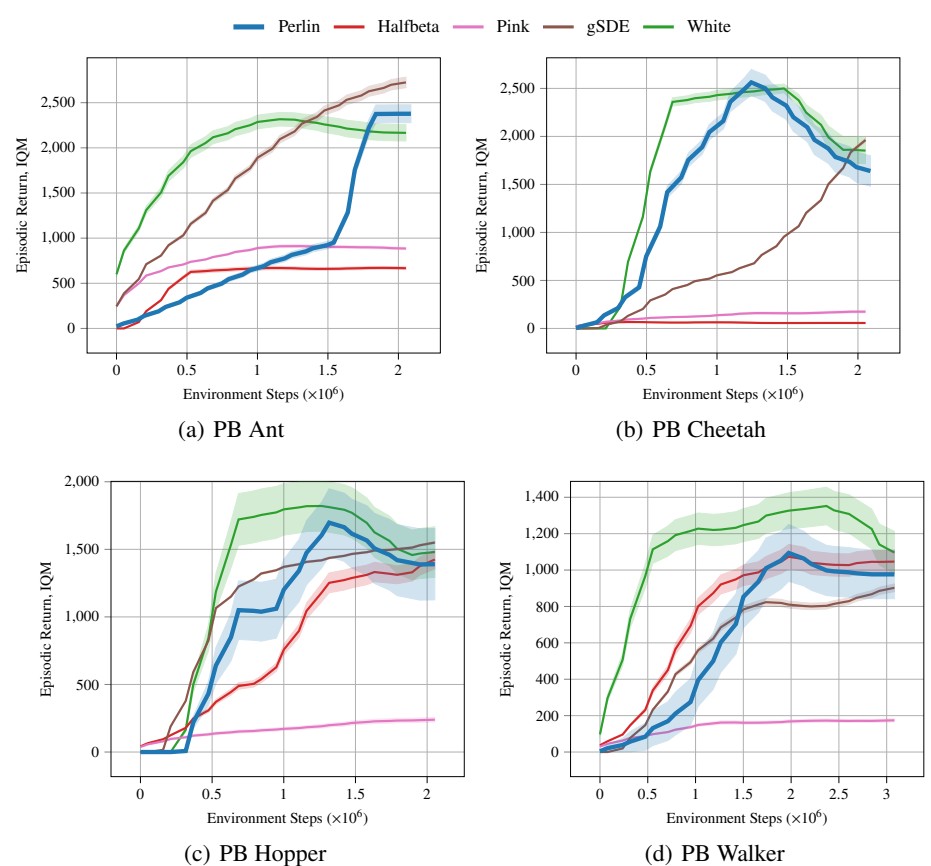

Figure 14: Results on selected environments from PyBullet.

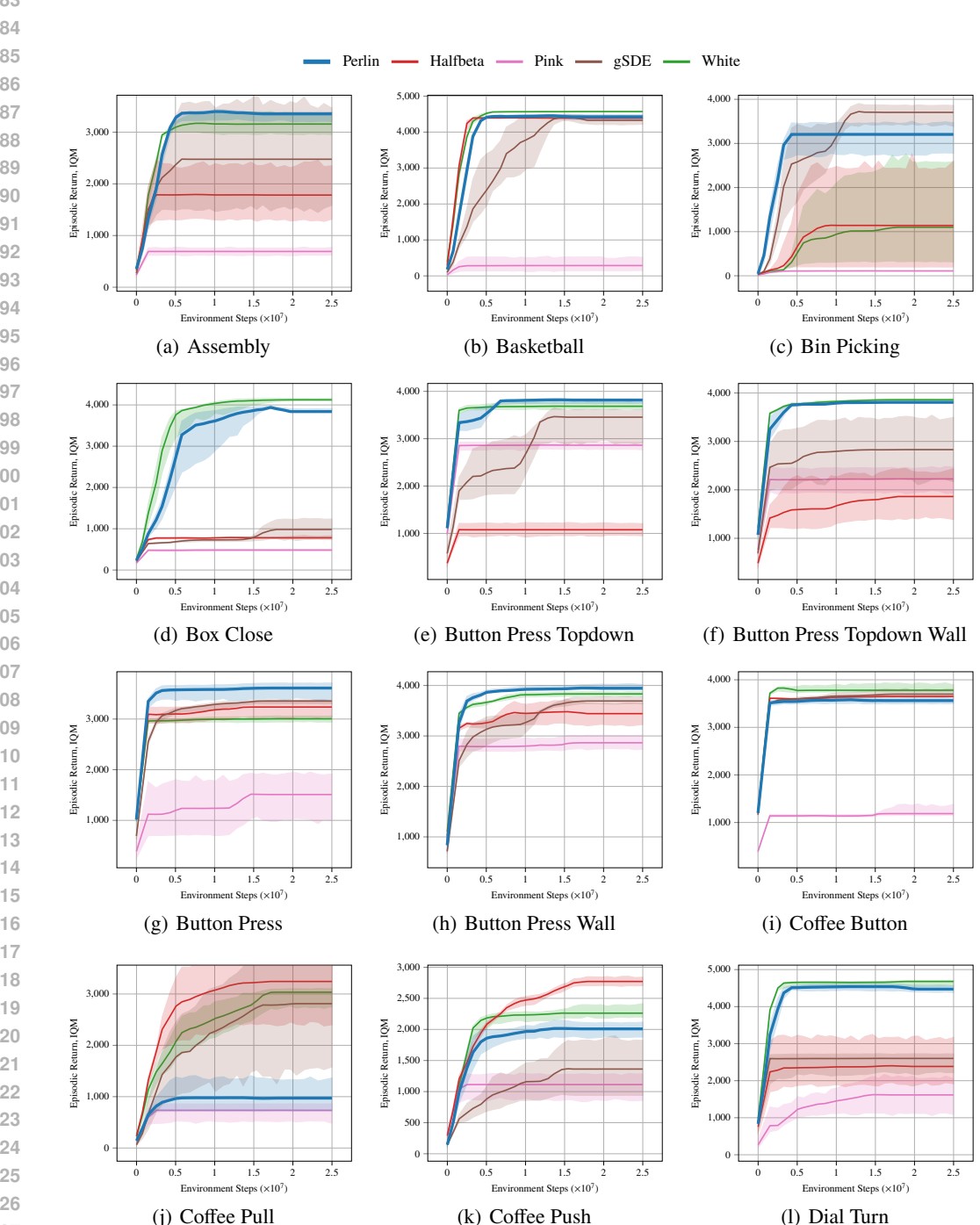

Figure 15: Results from all Metaworld (v2 variant) environments (Page 1).

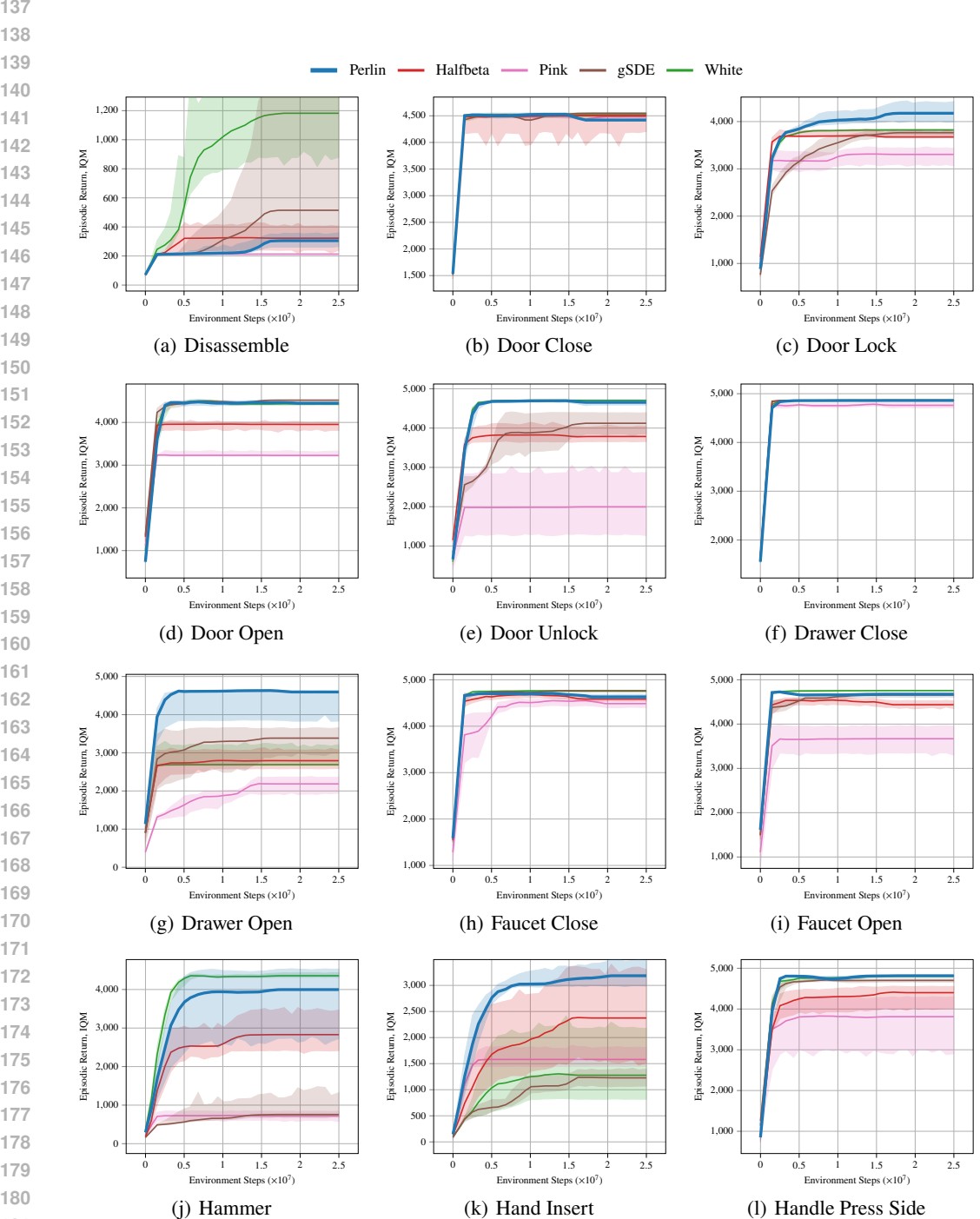

Figure 16: Results from all Metaworld (v2 variant) environments (Page 2).

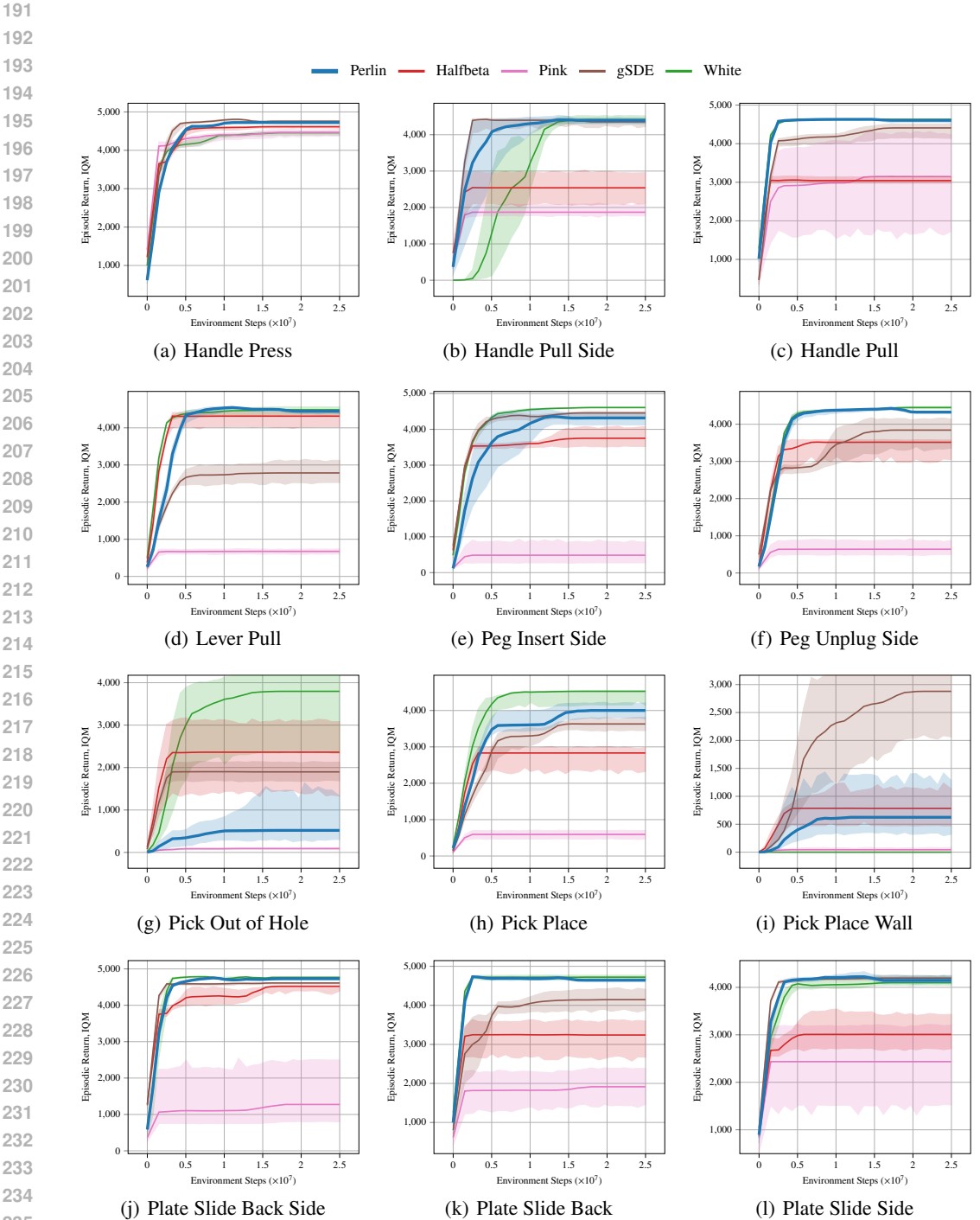

Figure 17: Results from all Metaworld (v2 variant) environments (Page 3).

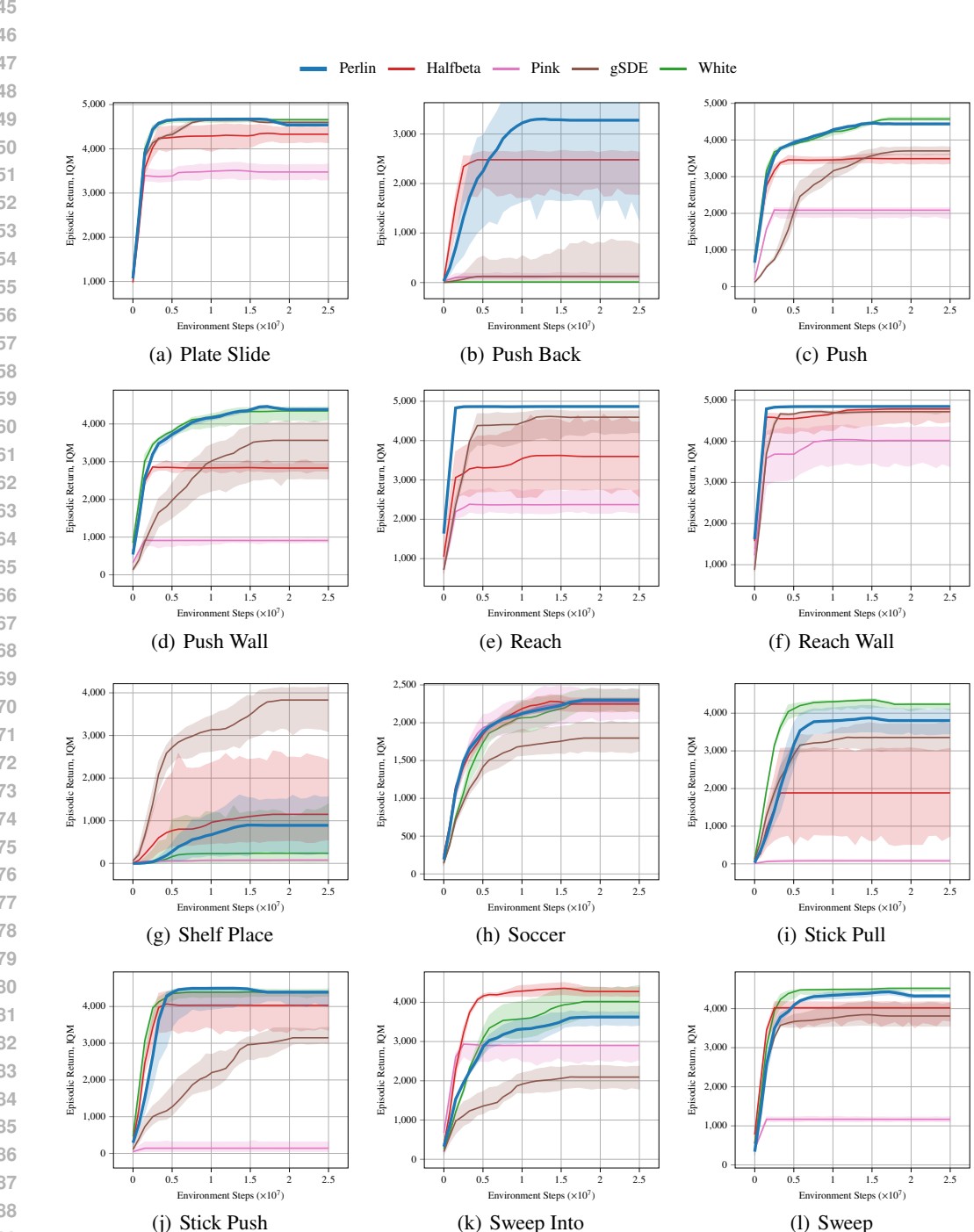

Figure 18: Results from all Metaworld (v2 variant) environments (Page 4).

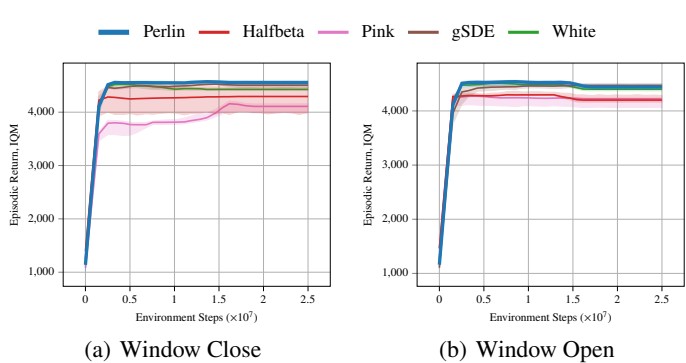

(a) Window Close    (b) Window Open

Figure 19: Results from all Metaworld (v2 variant) environments (Page 5).

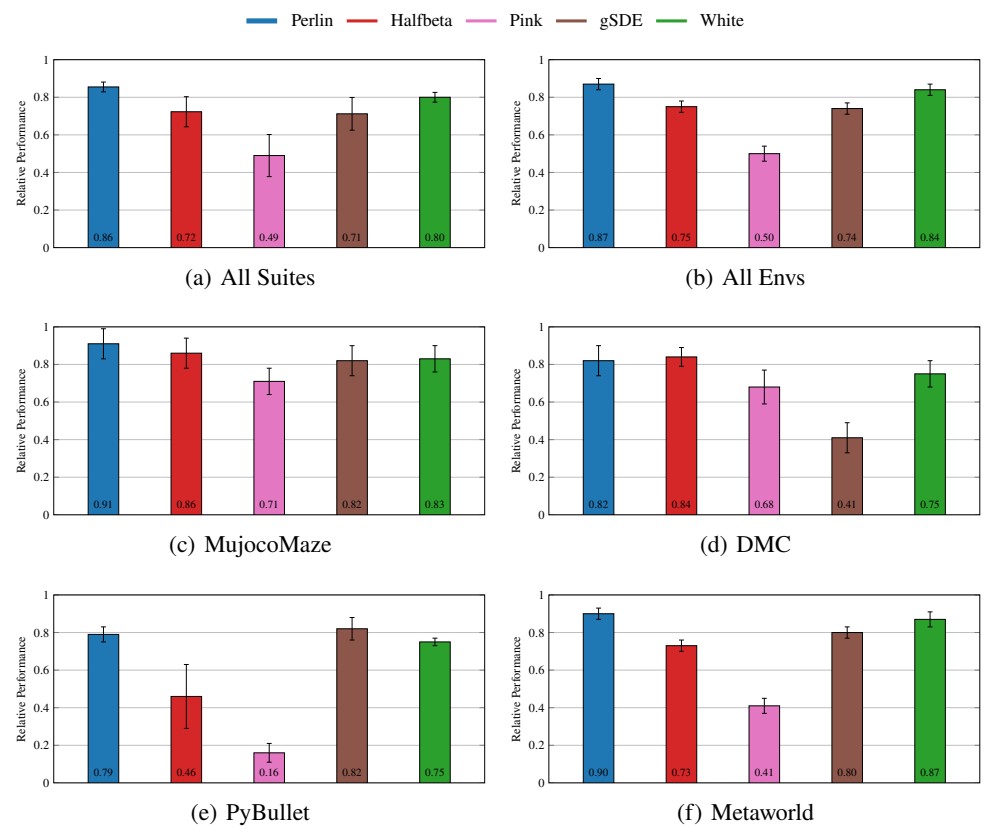

Figure 20: Aggregate results showing the mean and standard error (SE) of episodic reward across all environments from all suites, with performance for each environment normalized relative to its best-performing algorithm. Figure (a) is mean+SE over the results from all suites, excluding MountainCar and our custom mazes. Figure (b) shows the mean+SE over all environments, including MountainCar and our custom mazes.

# B SMOOTHNESS

## B.1 MEAN SQUARED JERK

Jerk is defined as the rate of change of acceleration with respect to time. The approximation of jerk depends on whether the actions represent speeds or accelerations. In both cases, jerk is numerically derived by estimating the change in acceleration over time.

When actions represent speeds, jerk is approximated by first calculating acceleration (the derivative of speed) and then computing the rate of change of acceleration. Given a discrete trajectory of speed actions $\{v_1, v_2, \ldots, v_T\}$, where $v_t$ represents the velocity at time $t$, the acceleration at time $t$ is given by:

$$a_t = \frac{v_{t+1} - v_t}{\Delta t}.$$

The jerk is then calculated as the change in acceleration between consecutive time steps:

$$j_t = \frac{a_{t+1} - a_t}{\Delta t}.$$

Thus, when actions are speeds, we need to compute both acceleration and jerk, requiring two numerical steps.

When actions represent accelerations, jerk is directly computed as the change in acceleration between consecutive time steps. For a discrete trajectory of acceleration actions $\{a_1, a_2, \ldots, a_T\}$, where $a_t$ is the acceleration at time $t$, the jerk is given by:

$$j_t = \frac{a_t - a_{t-1}}{\Delta t}.$$

In this case, jerk is calculated in a single step, as it directly measures the change in acceleration.

The Mean Squared Jerk (MSJ) is then calculated as the average of the squared jerk values across the trajectory:

$$\text{MSJ} = \frac{1}{T - 2} \sum_{t=1}^{T-2} j_t^2,$$

where $T$ is the total number of steps in the trajectory.

This metric provides a quantitative measure of smoothness for action sequences. Lower MSJ values indicate smoother trajectories, with less variation in the rate of acceleration. Whether the actions are speeds or accelerations, the MSJ helps assess the smoothness of exploration in reinforcement learning, as both the exploration noise and the learned policy dynamics influence the resulting jerk values.

## B.2 SMOTHNESS RESULTS

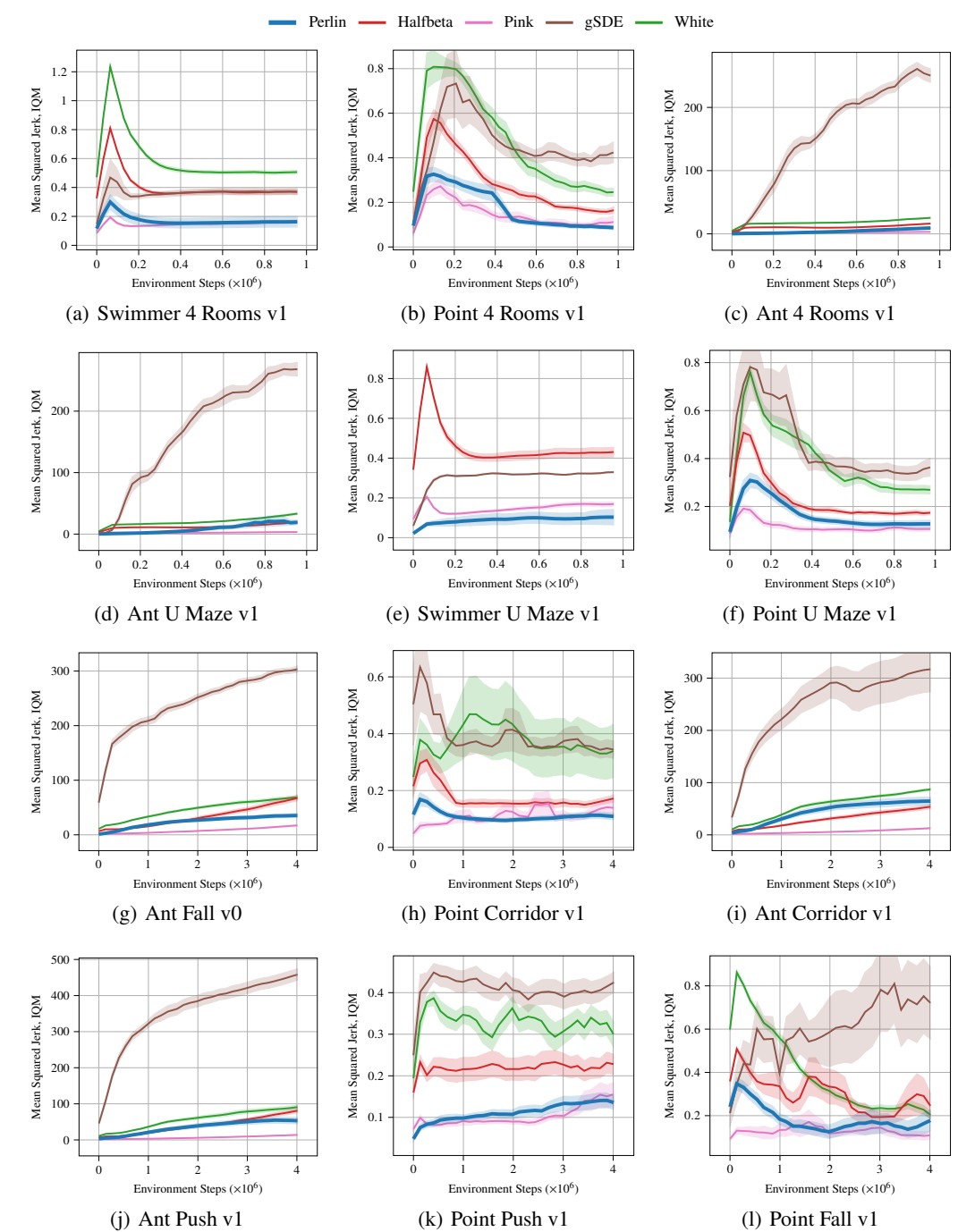

Figure 21: Smoothness (mean squared jerk, lower is better) on selected environments from MujocoMaze (Page 1).

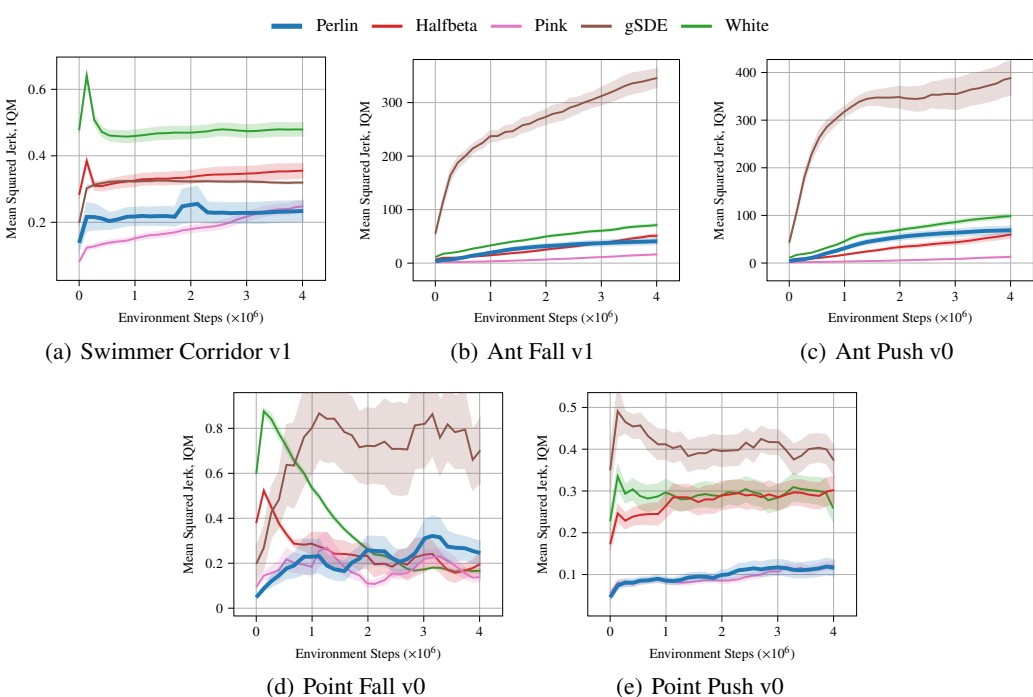

Figure 22: Smoothness (mean squared jerk, lower is better) on selected environments from MujocoMaze (Page 2).

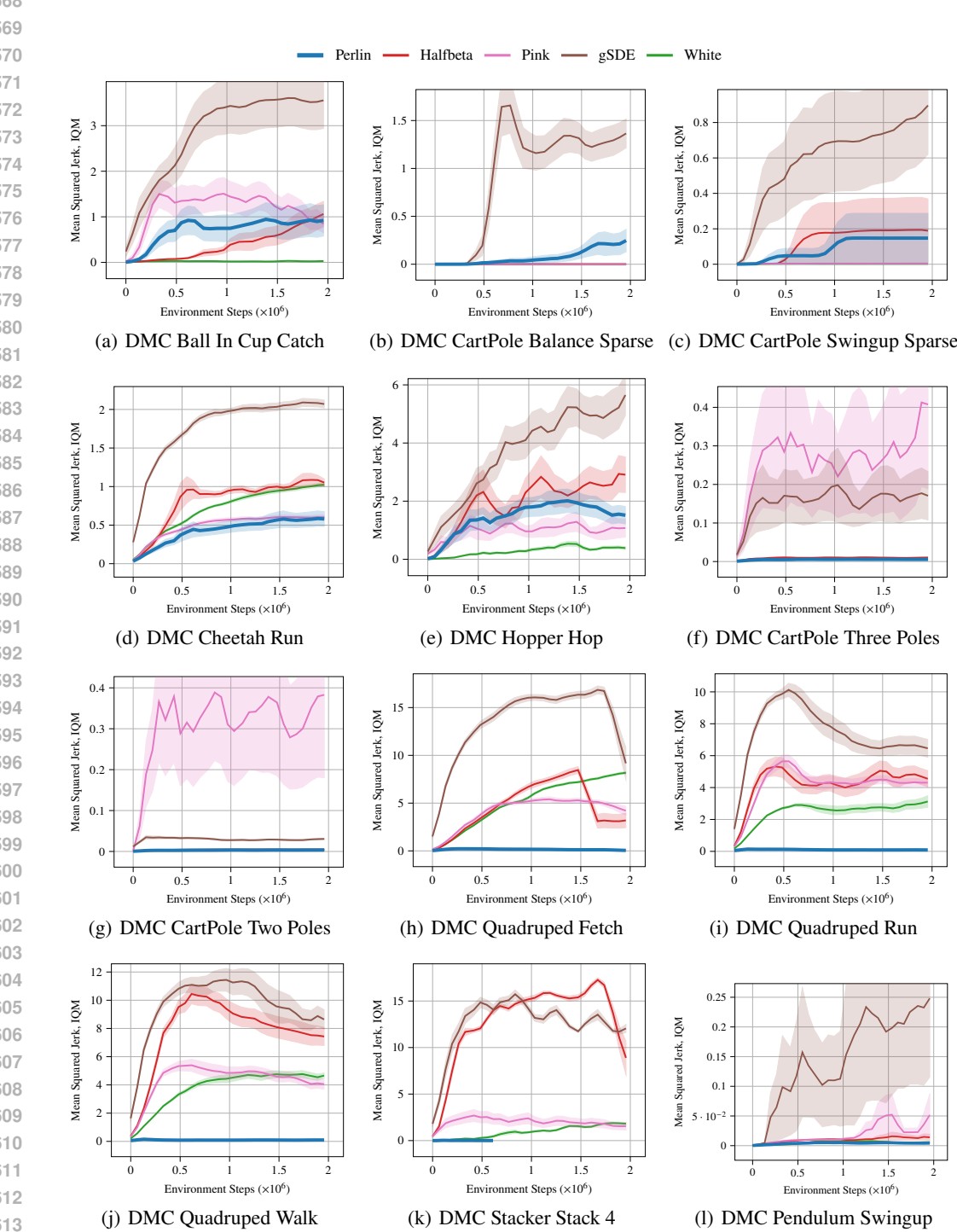

Figure 23: Smoothness (mean squared jerk, lower is better) on selected environments from DMC (Page 1).

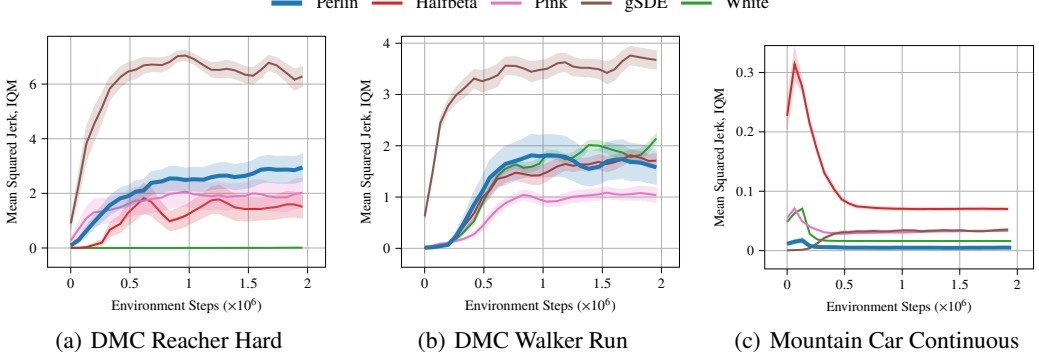

(a) DMC Reacher Hard  (b) DMC Walker Run  (c) Mountain Car Continuous

Figure 24: Smoothness (mean squared jerk, lower is better) on selected environments from DMC (Page 2) and MountainCarContinuous.

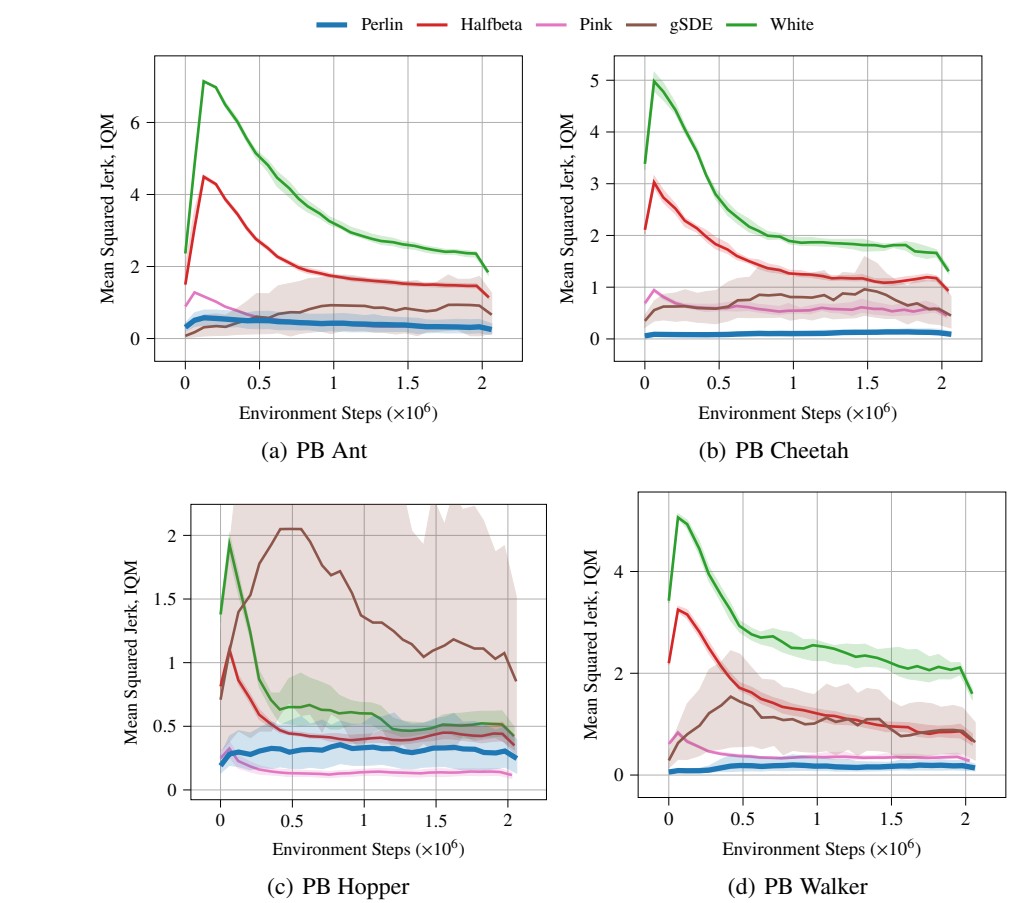

Figure 25: Smoothness (mean squared jerk, lower is better) on selected environments from PyBullet.

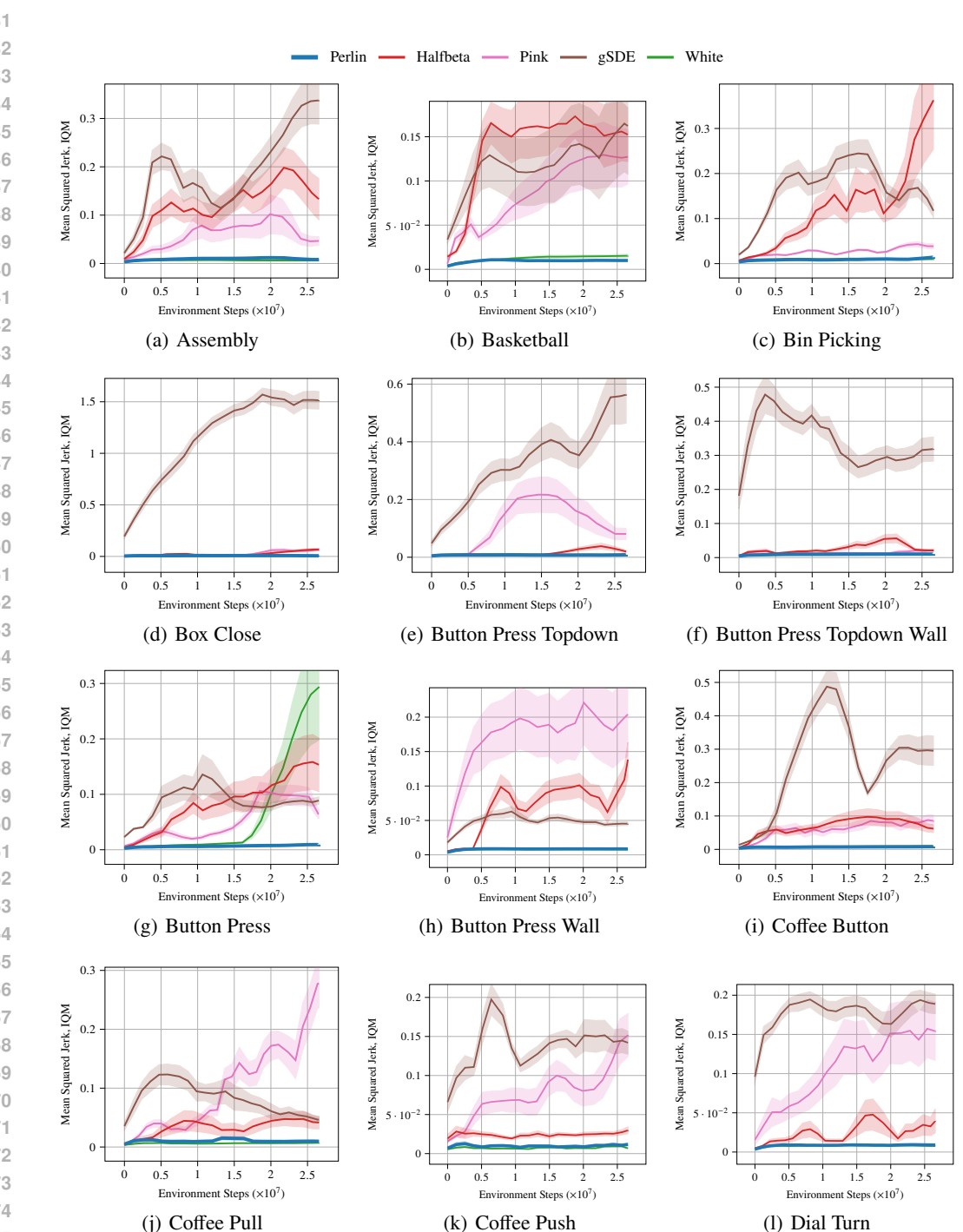

Figure 26: Smoothness (mean squared jerk, lower is better) from all Metaworld (v2 variant) environments (Page 1).

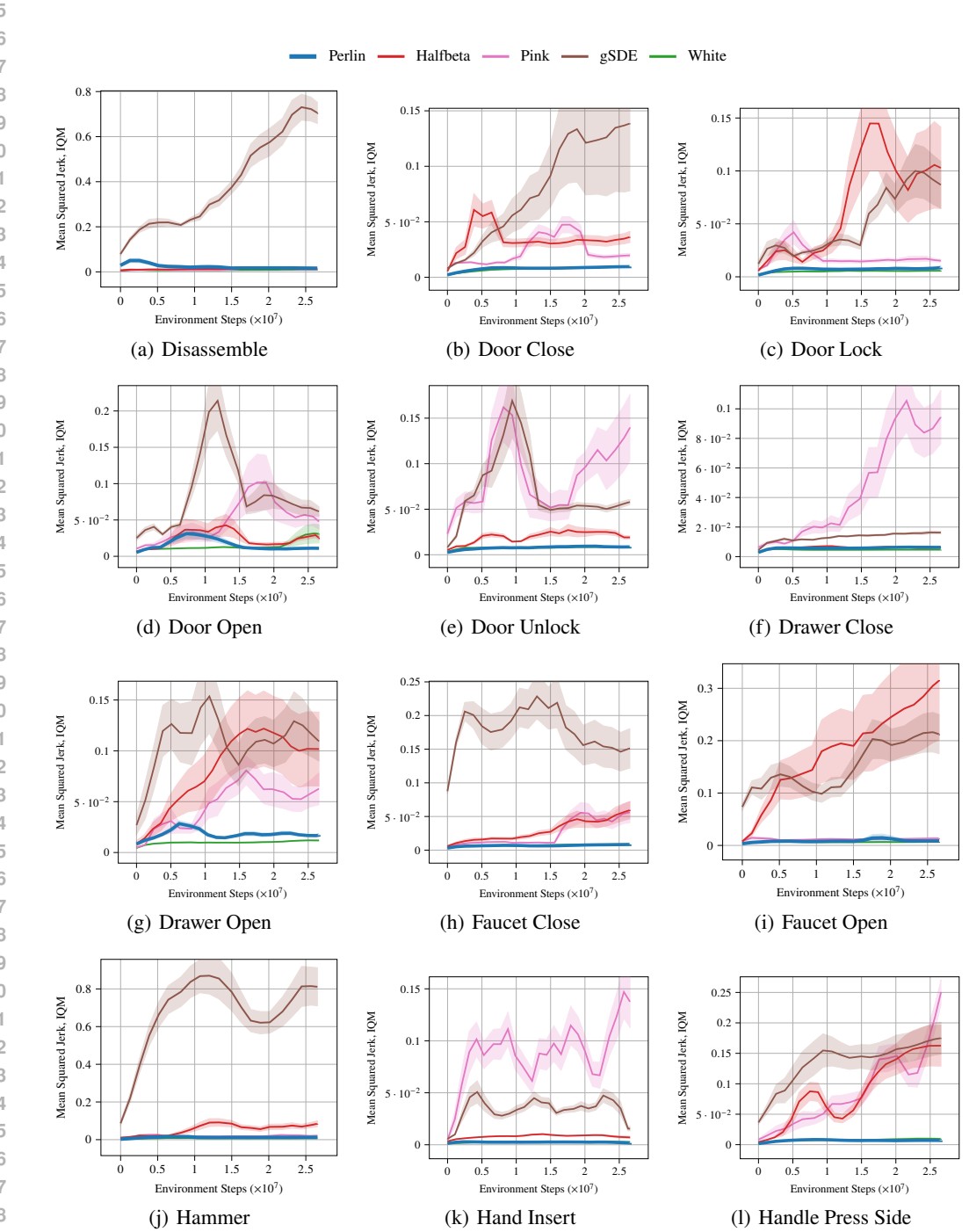

Figure 27: Smoothness (mean squared jerk, lower is better) from all Metaworld (v2 variant) environments (Page 2).

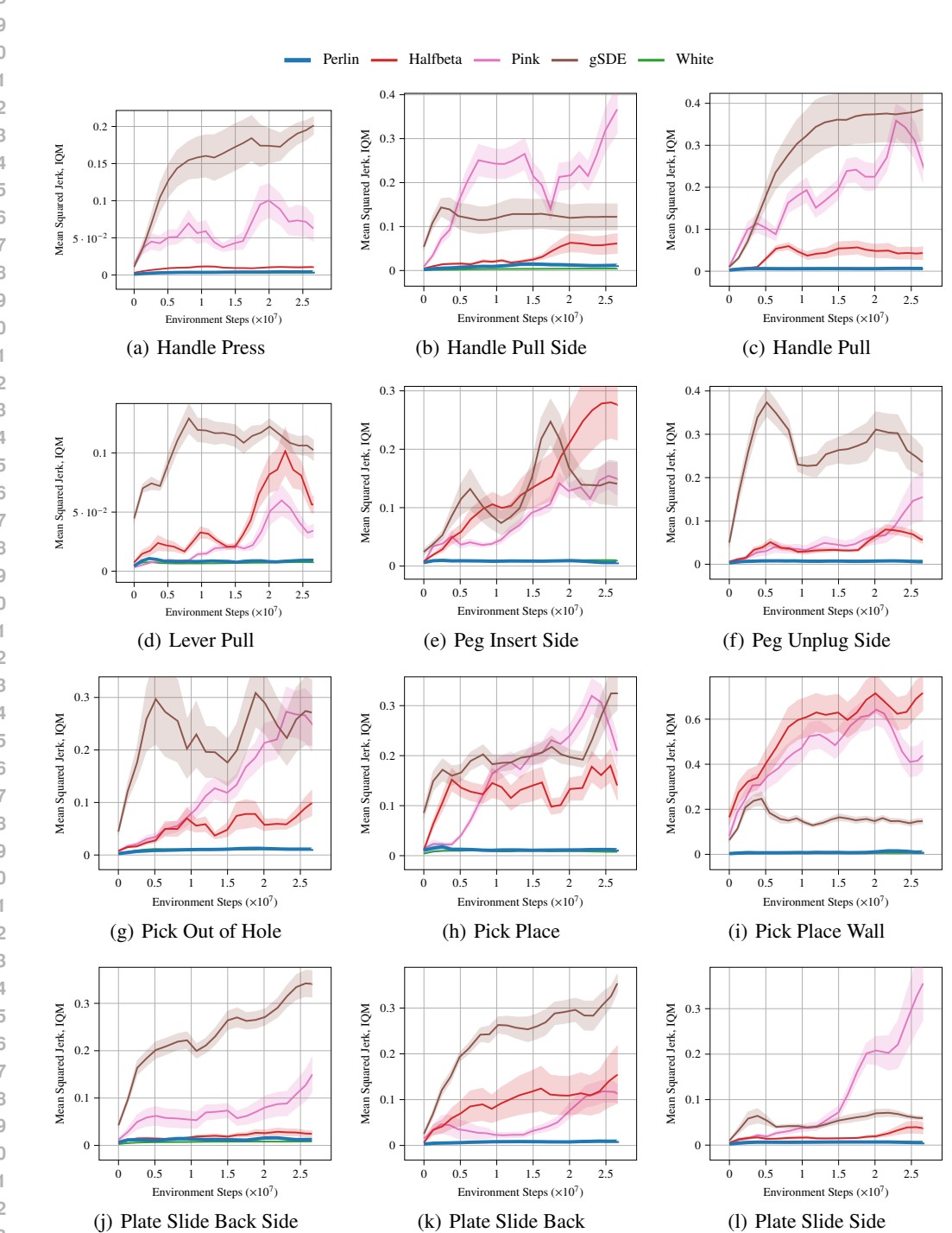

Figure 28: Smoothness (mean squared jerk, lower is better) from all Metaworld (v2 variant) environments (Page 3).

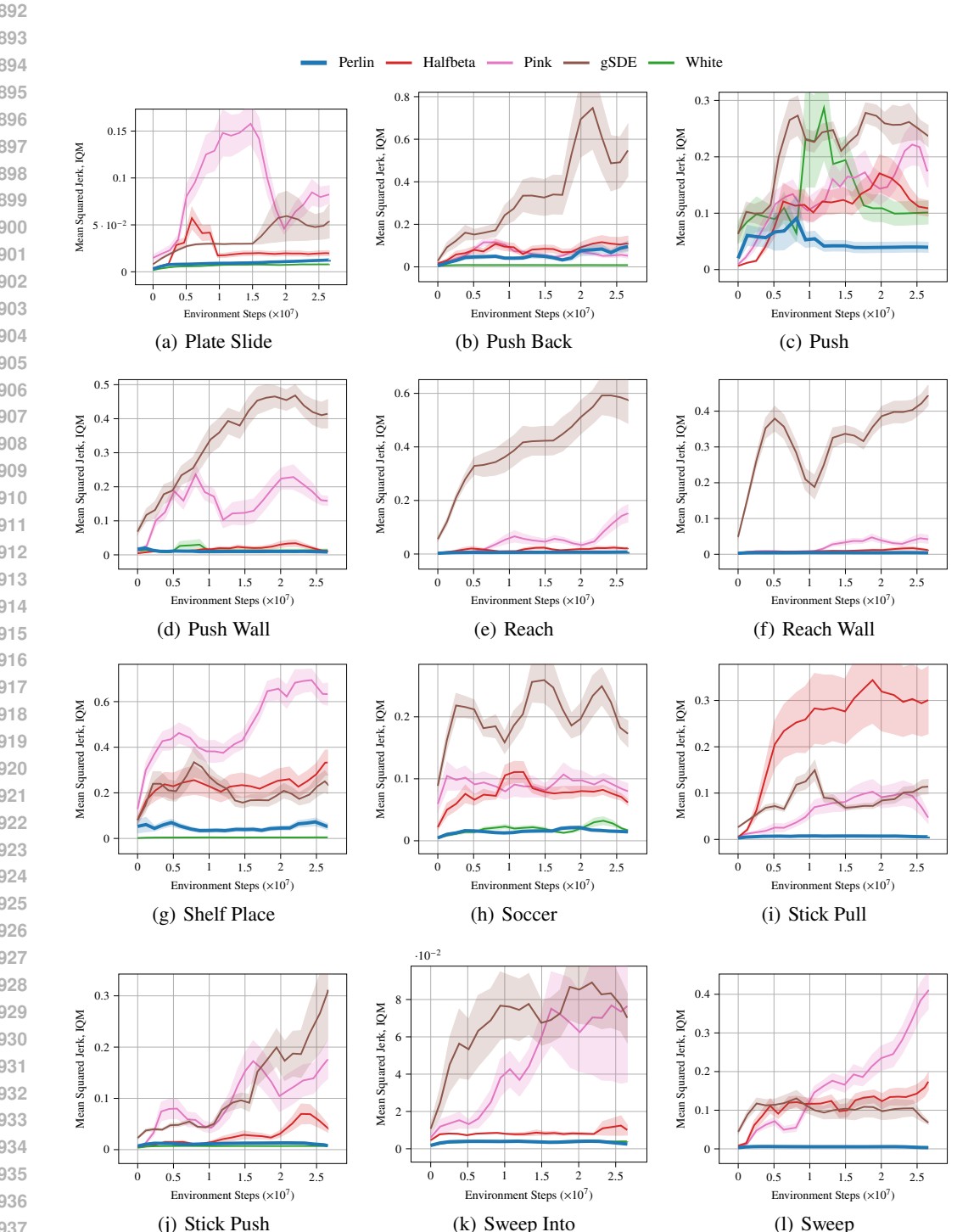

Figure 29: Smoothness (mean squared jerk, lower is better) from all Metaworld (v2 variant) environments (Page 4).

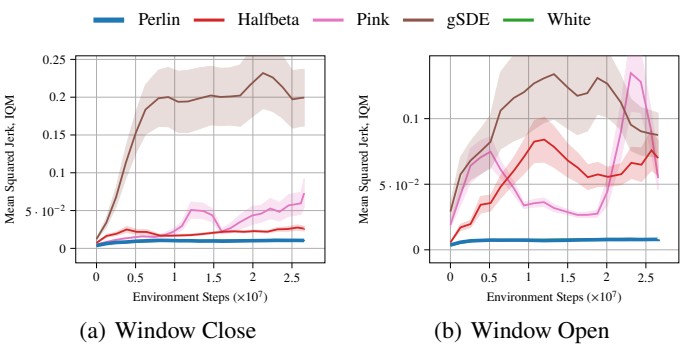

(a) Window Close   (b) Window Open

Figure 30: Smoothness (mean squared jerk, lower is better) from all Metaworld (v2 variant) environments (Page 5).

## C HPs

Table 1: Base HPs (used for MountainCar, Custom Mazes, MujocoMaze, DMC)

| | Perlin | White | gSDE | Pink | HalfBeta |
|---|---|---|---|---|---|
| num parallel envs | 4 | 4 | 4 | 4 | 4 |
| number samples (n_steps) | 2048 | 2048 | 2048 | 2048 | 2048 |
| GAE $\lambda$ | 0.95 | 0.95 | 0.95 | 0.95 | 0.95 |
| discount factor | 0.99 | 0.99 | 0.99 | 0.99 | 0.99 |
| | | | | | |
| optimizer | adam | adam | adam | adam | adam |
| epochs | 10 | 10 | 20 | 10 | 10 |
| learning rate | 2.5e-4 | 2.5e-4 | 2.5e-4 | 2.5e-4 | 2.5e-4 |
| use critic | True | True | True | True | True |
| epochs critic | 10 | 10 | 20 | 10 | 10 |
| learning rate critic | 2.5e-4 | 2.5e-4 | 3e-4 | 2.5e-4 | 2.5e-4 |
| batch size | 128 | 128 | 2048 | 128 | 128 |
| SDE sampling frequency (ssf) | n.a. | n.a. | 4 | n.a. | n.a. |
| $k_{speed}$ | 0.33 | n.a. | n.a. | n.a. | n.a. |
| $\beta$-coefficient | n.a. | 0 | n.a. | 1 | 0.5 |
| entropy coefficient | 0 | 0 | 0 | 0 | 0 |
| | | | | | |
| normalized observations | True | True | True | True | True |
| normalized rewards | True | True | True | True | True |
| PPO clip | 0.2 | 0.2 | 0.2 | 0.2 | 0.2 |
| max gradient norm | 0.5 | 0.5 | 0.5 | 0.5 | 0.5 |
| | | | | | |
| hidden layers | [64, 64] | [64, 64] | [64, 64] | [64, 64] | [64, 64] |
| hidden layers critic | [64, 64] | [64, 64] | [64, 64] | [64, 64] | [64, 64] |
| hidden activation | tanh | tanh | tanh | tanh | tanh |
| initial std | 1.0 | 1.0 | 1.0 | 1.0 | 1.0 |

Table 2: PyBullet HPs

| | Perlin | White | gSDE | Pink | HalfBeta |
|---|---|---|---|---|---|
| num parallel envs | 4 | 4 | 4 | 4 | 4 |
| number samples (n_steps) | 8192 | 8192 | 8192 | 8192 | 8192 |
| GAE $\lambda$ | 0.9 | 0.9 | 0.9 | 0.9 | 0.9 |
| discount factor | 0.99 | 0.99 | 0.99 | 0.99 | 0.99 |
| | | | | | |
| optimizer | adam | adam | adam | adam | adam |
| epochs | 20 | 20 | 20 | 20 | 20 |
| learning rate | 3e-4 | 3e-4 | 3e-4 | 3e-4 | 3e-4 |
| use critic | True | True | True | True | True |
| epochs critic | 20 | 20 | 20 | 20 | 20 |
| learning rate critic | 3e-4 | 3e-4 | 3e-4 | 3e-4 | 3e-4 |
| batch size | 128 | 128 | 128 | 128 | 128 |
| SDE sampling frequency (ssf) | n.a. | n.a. | n.a. | n.a. | n.a. |
| $k_{speed}$ | 0.33 | 0.33 | 0.33 | 0.33 | 0.33 |
| $\beta$-coefficient | n.a. | 0 | n.a. | 1 | 0.5 |
| entropy coefficient | 0 | 0 | 0 | 0 | 0 |
| | | | | | |
| normalized observations | True | True | True | True | True |
| normalized rewards | True | True | True | True | True |
| PPO clip | 0.4 | 0.4 | 0.4 | 0.4 | 0.4 |
| max gradient norm | 0.5 | 0.5 | 0.5 | 0.5 | 0.5 |
| | | | | | |
| hidden layers | [64, 64] | [64, 64] | [64, 64] | [64, 64] | [64, 64] |
| hidden layers critic | [64, 64] | [64, 64] | [64, 64] | [64, 64] | [64, 64] |
| hidden activation | tanh | tanh | tanh | tanh | tanh |
| initial std | 0.33 | 0.33 | 0.33 | 0.33 | 0.33 |

Table 3: Metaworld HPs

|  | Perlin | White | gSDE | Pink | HalfBeta |
|---|---|---|---|---|---|
| num parallel envs | 4 | 4 | 4 | 4 | 4 |
| number samples (n_steps) | 16000 | 16000 | 16000 | 16000 | 16000 |
| GAE $\lambda$ | 0.95 | 0.95 | 0.95 | 0.95 | 0.95 |
| discount factor | 0.99 | 0.99 | 0.99 | 0.99 | 0.99 |
| optimizer | adam | adam | adam | adam | adam |
| epochs | 10 | 10 | 10 | 10 | 10 |
| learning rate | 1e-3 | 1e-3 | 1e-3 | 1e-3 | 1e-3 |
| use critic | True | True | True | True | True |
| epochs critic | 10 | 10 | 10 | 10 | 10 |
| learning rate critic | 1e-3 | 1e-3 | 1e-3 | 1e-3 | 1e-3 |
| batch size | 500 | 500 | 500 | 500 | 500 |
| SDE sampling frequency (ssf) | n.a. | n.a. | n.a. | n.a. | n.a. |
| $k_{speed}$ | 0.33 | 0.33 | 0.33 | 0.33 | 0.33 |
| $\beta$-coefficient | n.a. | 0 | n.a. | 1 | 0.5 |
| entropy coefficient | 0 | 0 | 0 | 0 | 0 |
| normalized observations | True | True | True | True | True |
| normalized rewards | True | True | True | True | True |
| PPO clip | lin* | lin* | lin* | lin* | lin* |
| max gradient norm | 0.5 | 0.5 | 0.5 | 0.5 | 0.5 |
| hidden layers | [64, 64] | [64, 64] | [64, 64] | [64, 64] | [64, 64] |
| hidden layers critic | [64, 64] | [64, 64] | [64, 64] | [64, 64] | [64, 64] |
| hidden activation | tanh | tanh | tanh | tanh | tanh |
| orthogonal initialization | True | True | True | True | True |
| initial std | 1.0 | 1.0 | 1.0 | 1.0 | 1.0 |

*Linear schedule from 0.25 to 0.01 during first 2/3 of training, then continued with 0.01.

# D  DERIVATIONS

## D.1  PERLIN NOISE HAS ZERO MEAN

We aim to show that Perlin noise has a zero mean. Let $\text{AntiPerlin}(x)$ denote the inverse of Perlin noise, defined as:

$$\text{AntiPerlin}(x) = -\text{Perlin}(x).$$

### D.1.1  CONSTRUCTION OF ANTIPERLIN FROM PERLIN

The Perlin noise function is constructed by generating unit-length gradient vectors at lattice points and interpolating between them. To construct $\text{AntiPerlin}(x)$, we can simply flip the sign of all gradient vectors used in the construction of Perlin noise:

$$\mathbf{g_i}^{\text{Anti}} = -\mathbf{g_i},$$

where $\mathbf{g_i}$ is a gradient vector at lattice point $\mathbf{i}$.

This transformation yields $\text{AntiPerlin}(x) = -\text{Perlin}(x)$, since flipping the sign of all gradients results in the negation of the entire noise function.

### D.1.2  EQUIVALENCE OF DISTRIBUTIONS

Perlin noise gradients are uniformly sampled from a distribution $P(g)$. Therefore, the probability of sampling a gradient $g$ is equal to the probability of sampling $-g$:

$$P(g) = P(-g).$$

As a result, the distribution of gradients used to generate $\text{Perlin}(x)$ is identical to that used for $\text{AntiPerlin}(x)$. Thus, $\text{AntiPerlin}(x)$ follows the same distribution as $\text{Perlin}(x)$ under random seeds of the PRNG.

### D.1.3  CONCLUSION

Since $\text{AntiPerlin} = -\text{Perlin}$ and both functions are equal in distribution, it must be that

$$\mathbb{E}[\text{Perlin}] = \mathbb{E}[-\text{Perlin}],$$

which implies that Perlin noise is symmetric around zero and has a zero mean.

## D.2  FINITE MOMENTS OF PERLIN NOISE

### D.2.1  BOUNDEDNESS OF PERLIN NOISE

Perlin noise is generated by summing weighted dot products between unit-length gradient vectors and displacements from lattice points in a hypercube. Given a point $\mathbf{x} \in \mathbb{R}^n$, the Perlin noise value is computed as:

$$\text{Perlin}(\mathbf{x}) = \sum_{\mathbf{i} \in \mathbf{I}} \left( \mathbf{d_i} \prod_{k=1}^{n} \Phi(x_k - i_k) \right),$$

where: $\mathbf{I}$ is the set of $2^n$ surrounding lattice points, $\mathbf{d_i} = \mathbf{g_i} \cdot (\mathbf{x} - \mathbf{i})$ is the dot product between the unit gradient vector $\mathbf{g_i}$ and the displacement $\mathbf{x} - \mathbf{i}$, $\Phi(t) = 3t^2 - 2t^3$ is the smooth interpolation function.

### D.2.2  BOUNDEDNESS OF INDIVIDUAL COMPONENTS

**Dot Products**: Since the gradient vectors $\mathbf{g_i}$ are unit-length ($\|\mathbf{g_i}\| = 1$) and each component of the displacement $(x_k - i_k)$ lies in the interval $[0, 1]$, each dot product $\mathbf{d_i}$ is bounded:

$$|\mathbf{d_i}| \leq D,$$

where $D = \sqrt{n}$, the maximum value when the displacement vector is $(1, 1, \ldots, 1)$ and the gradient vector is aligned with the displacement.

**Interpolation Function**: The interpolation function $\Phi(t)$ satisfies $0 \leq \Phi(t) \leq 1$ for $t \in [0, 1]$. Therefore, the product $\prod_{k=1}^{n} \Phi(x_k - i_k)$ is also bounded between 0 and 1.

### D.2.3 Boundedness of Perlin Noise Value

Each term in the Perlin noise sum is the product of a bounded dot product and a bounded interpolation factor

$$\left| \mathbf{d_i} \prod_{k=1}^{n} \Phi(x_k - i_k) \right| \leq D.$$

Since there are $2^n$ terms in the sum (one for each surrounding lattice point), the Perlin noise value is bounded by

$$|\text{Perlin}(\mathbf{x})| \leq C,$$

where $C = 2^n D$.

### D.2.4 Finite Moments

For a bounded random variable $X$ with $|X| \leq C$, all moments of any order $k \geq 1$ are finite:

$$E[|X|^k] \leq C^k.$$

Applying this to the Perlin noise:

$$E[|\text{Perlin}(\mathbf{x})|^k] \leq C^k,$$

which confirms that all moments of the Perlin noise are finite.

### D.2.5 Conclusion

The bounded nature of the Perlin noise function ensures that all its moments, regardless of order, are finite. This holds true due to the bounded dot products, the bounded interpolation function, and the finite number of terms in the summation. Thus, we conclude that the Perlin noise has finite moments of all orders.

## D.3 Analyticity of the Normalization Function

We aim to show that the normalization function $N(x)$, which transforms Perlin noise into a Gaussian-like distribution, is analytic. An analytic function is one that is infinitely differentiable and can be represented as a polynomial expansion around a point $x_0$.

### D.3.1 Smoothness of Perlin Noise

Perlin noise is generated using smooth gradient functions. As these gradients are continuous and differentiable, the Perlin noise function $P(x, y)$ inherits these properties, making it smooth and infinitely differentiable.

### D.3.2 Construction of the Normalization Function

The normalization function $N(x)$ maps Perlin noise values to a standard Gaussian-like distribution. If $F(x)$ is the cumulative distribution function (CDF) of Perlin noise, we can express $N(x)$ using the inverse CDF of the standard normal distribution $\Phi^{-1}$:

$$N(x) = \Phi^{-1}(F(x)).$$

### D.3.3 Smoothness of the Transformation

Since the CDF $F(x)$ of Perlin noise is smooth and the inverse Gaussian CDF $\Phi^{-1}(x)$ is smooth, their composition $N(x) = \Phi^{-1}(F(x))$ is also smooth and infinitely differentiable.

### D.3.4 Conclusion

Because $N(x)$ is infinitely differentiable, it is analytic and can thus be expressed as a polynomial expansion.

# E    ADDITIONAL FIGURES

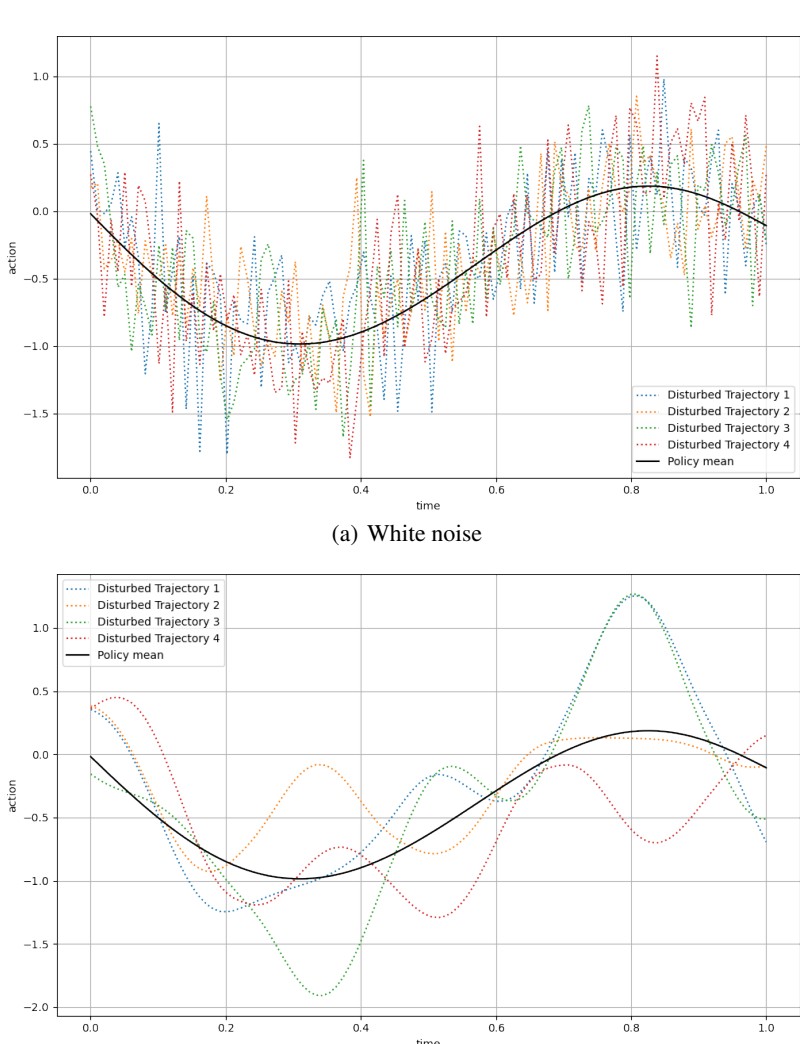

(a) White noise

(b) Perlin noise

Figure 31: Comparison of Sampled Trajectories via White noise or Perlin noise.

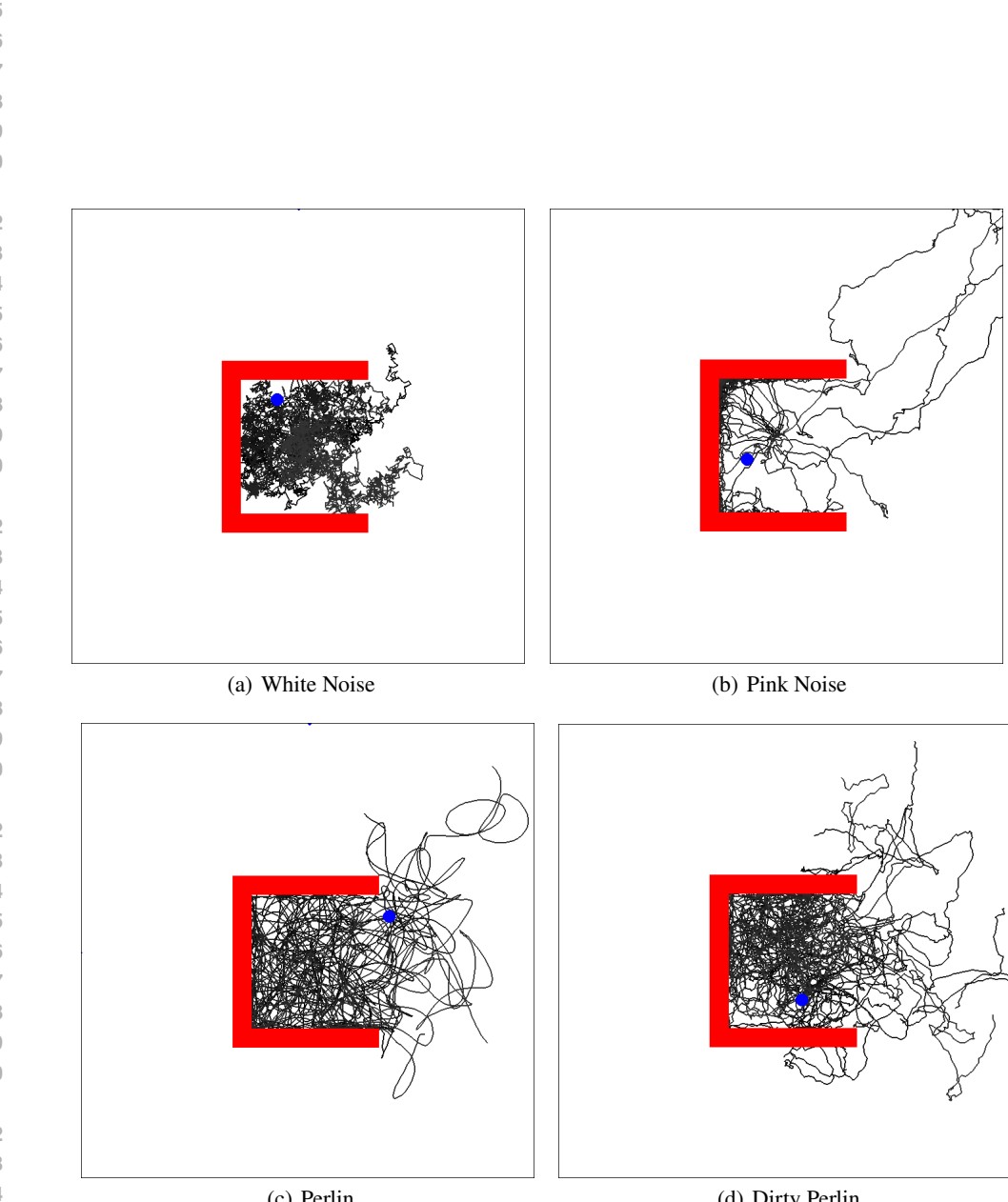

(a) White Noise

(b) Pink Noise

(c) Perlin

(d) Dirty Perlin

Figure 32: Particles driven by noise 'exploring' a box.

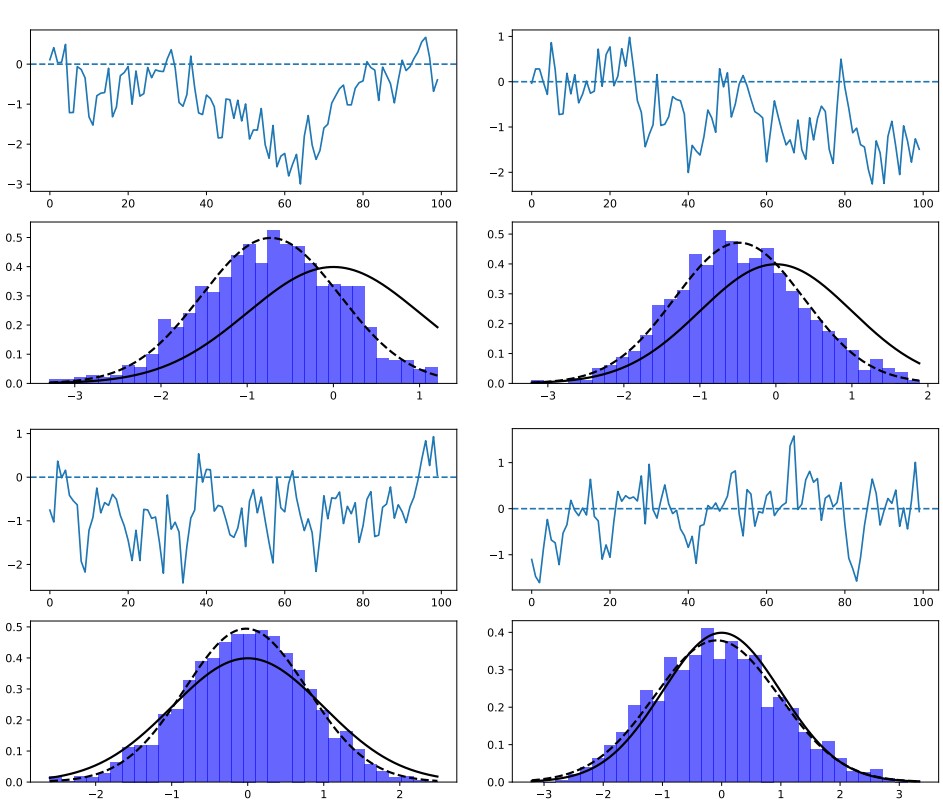

Figure 33: Multiple simulations for Pink noise.

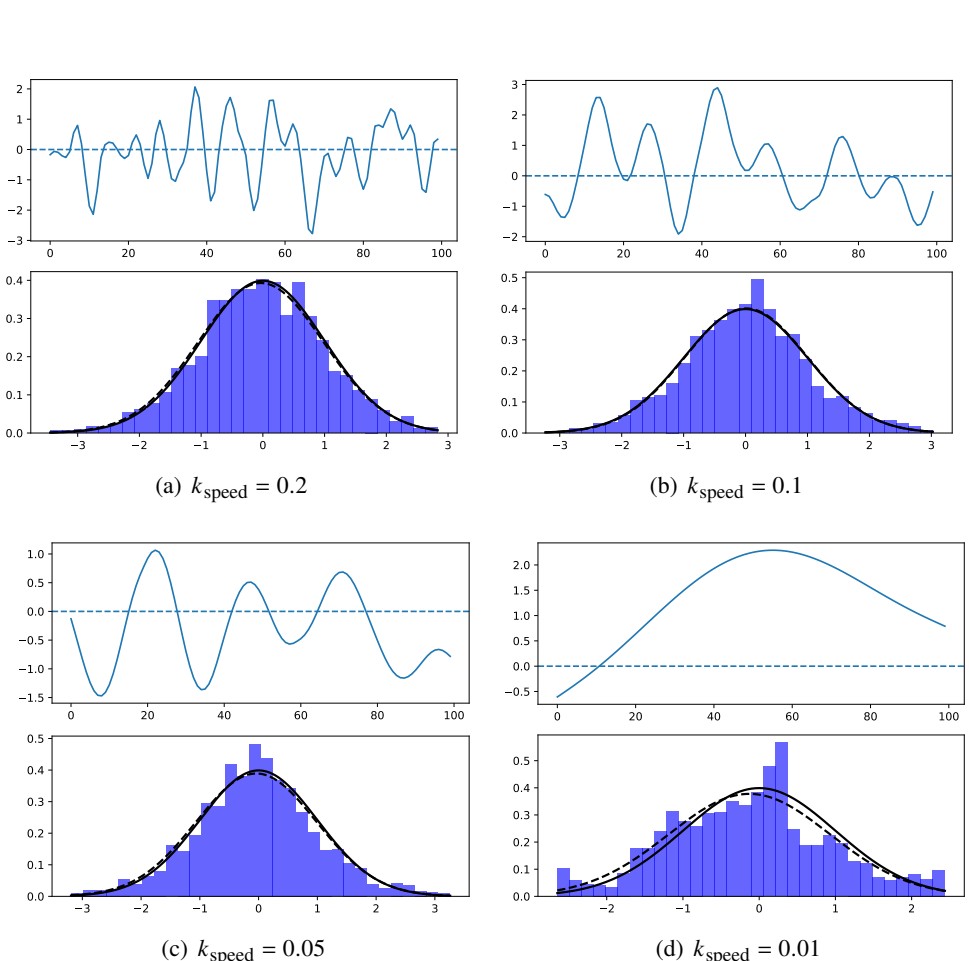

Figure 34: Perlin noise with different $k_{\text{speed}}$ settings.

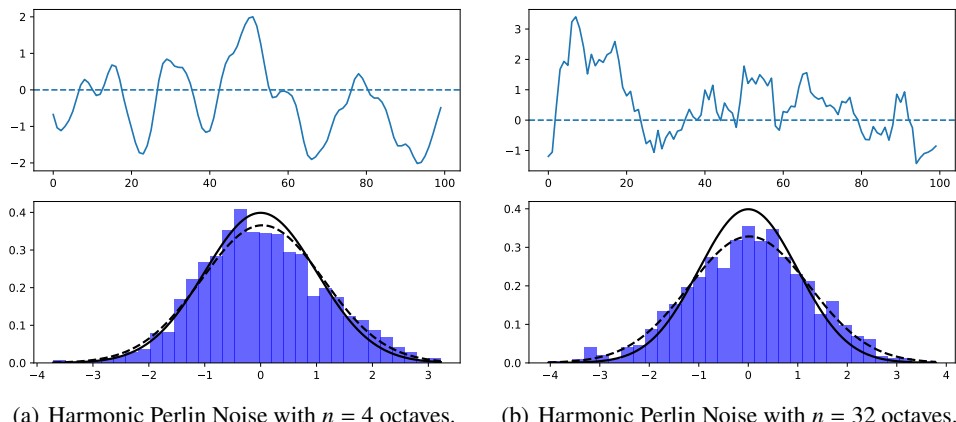

(a) Harmonic Perlin Noise with $n = 4$ octaves. (b) Harmonic Perlin Noise with $n = 32$ octaves.

Figure 35: Harmonic Perlin with $k_{\text{speed}} = 0.05$.

## F FURTHER COMPOSITIONAL NOISES FOR RL

Harmonic Perlin is constructed by superimposing multiple Perlin noise functions with different frequencies and amplitudes. This allows us to construct fractal noises, that have more complex noise pattern and have richer substructures while trading away some of the smoothness. We describe the individual Perlin noises as octaves. While potentially any kind of mixture based on different frequencies and amplitudes is possible, we can reduce the number of additional hyperparameters by enforcing the relation between these to follow the harmonic series. As such Harmonic Perlin will be described by (formula shown is missing a correction term to ensure $\sigma = 1$)

$$\text{HarmonicPerlin}(x) = \frac{\sum_{i=1}^{n} \frac{1}{2^i} \mathcal{P}_i(x)}{\sum_{i=1}^{n} \frac{1}{2^i}},$$

where $k_{\text{speed}}$ for $\mathcal{P}_i$ is set to $2^n \cdot k_{\text{speed}}$.

As a way to efficiently approximate Harmonic Perlin with a large number of octaves we propose Dirty Perlin by defining

$$\text{DirtyPerlin}(x) = \frac{f\epsilon + (1 - f)\mathcal{P}(x)}{\sqrt{f^2 + (1 - f)^2}},$$

where $\epsilon \sim \mathcal{N}(0, 1)$ and $f = k_{dirty\_ratio}$.

We can use these two noises for exploration in the same fashion as already described for Perlin noise. While these compositions allow us to design elaborate exploration noises with desired properties, we must also question whether this is a good idea. Is a method that achieves better performance than another, while requiring more HPs, actually better? Or are we just shiting the work of solving the task from the RL algorithm to the researcher in charge of HP tuning? In our view, every added hyperparameter increases the risk of overfitting them to the tasks, making it challenging to determine whether the ML algorithm truly represents an improvement on its own.

For low numbers of octaves we observe the desired smooth substructures overlayed with the Perlin of the first harmonic (can be seen in Figure 35 (a)). Higher octaves lead to less smooth trajectories. High number of octaves (can be seen in Figure 35 (b)) behave similar to Pink noise, in that we observe long terms trends, while generating unsmooth trajectories. Contrary to Pink noise, our samples remain to be on-policy. The empirical parameters remain close to the policy parameters.

Dirty Perlin (Figure 36) behaves similar to high octave Harmonic Perlin, while being a lot cheaper computationally and can therefore be used as an approximation of high octave Harmonic Perlin.

Figure 32 shows the exploration behavior in 2D as a combination the general behavior of Perlin with unsmooth substructures.

We present these additional noise methods as they may be of interest to the reader. While preliminary tests did not show statistically significant overperformance, further experiments were not conducted. We also see the danger of bloating the exploration method with unnecessary complexity and additional HPs. The codebase we provide includes implementations for testing these noise types, allowing for easy experimentation.

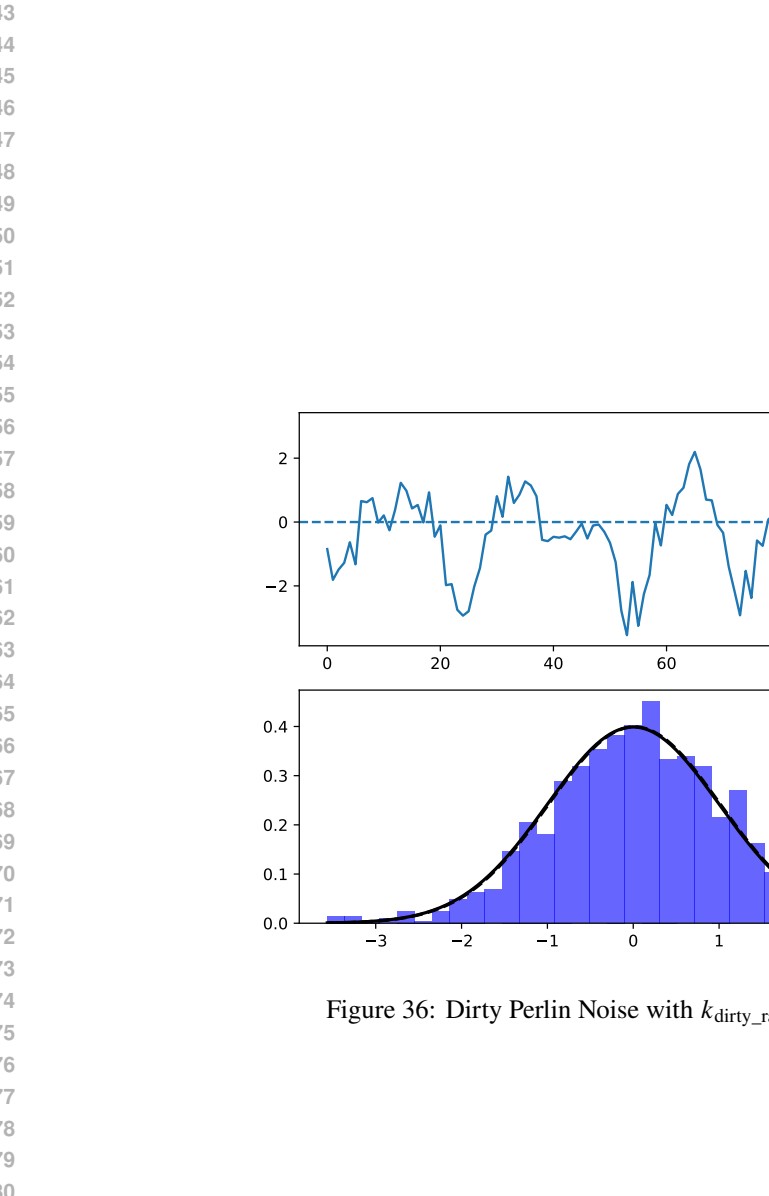

Figure 36: Dirty Perlin Noise with $k_{\text{dirty\_ratio}} = \frac{1}{3}$.

# G  PSEUDOCODE

```python
import math

def interpolate(a0, a1, t):
    # Smoothstep interpolation
    w = t * t * (3 - 2 * t)
    return (a1 - a0) * w + a0

def random_gradient(ix, iy):
    # Generate a pseudo-random angle based on coordinates
    angle = hash((ix, iy)) % (2 * math.pi)
    return (math.cos(angle), math.sin(angle))

def dot_grid_gradient(gradients, ix, iy, x, y):
    gradient = gradients[(ix, iy)]
    dx, dy = x - ix, y - iy
    return dx * gradient[0] + dy * gradient[1]

def perlin(x, y):
    # Determine grid cell coordinates
    x0, x1 = math.floor(x), math.floor(x) + 1
    y0, y1 = math.floor(y), math.floor(y) + 1

    # Precompute gradients for the grid points
    gradients = {
        (x0, y0): random_gradient(x0, y0),
        (x1, y0): random_gradient(x1, y0),
        (x0, y1): random_gradient(x0, y1),
        (x1, y1): random_gradient(x1, y1)
    }

    # Determine interpolation weights
    sx, sy = x - x0, y - y0

    # Compute dot product at each grid point
    v00 = dot_grid_gradient(gradients, x0, y0, x, y)
    v10 = dot_grid_gradient(gradients, x1, y0, x, y)
    v01 = dot_grid_gradient(gradients, x0, y1, x, y)
    v11 = dot_grid_gradient(gradients, x1, y1, x, y)

    # Perform smoothstep interpolation
    i1 = interpolate(v00, v10, sx)
    i2 = interpolate(v01, v11, sx)

    # Final smoothstep interpolation in the y dimension
    return interpolate(i1, i2, sy)
```

Listing 1: 2D Perlin Noise 'Pseudocode' (actually valid Python)

