# OpenReview forum: "Perlin Noise for Exploration in Reinforcement Learning"
_ICLR.cc/2025/Conference — Submitted to ICLR 2025_

### Official Review · Reviewer_nqa9 · 2024-10-28

**Soundness:** 2
**Presentation:** 2
**Contribution:** 3
**Rating:** 5
**Confidence:** 3

**Summary:**

This paper proposes a new exploration strategy derived from Perlin Noise for Reinforcement Learning tasks. Instead of using classical Gaussian Noise, the authors found that Perlin noise could explore the hard-exploration task more effectively and smoothly. To allow Perlin noise to be used for training the RL algorithms, the authors conduct a theoretical work which shows the asymtotic normality of Perlin noise.  A normalization function is also proposed to map the Perlin noises to be gaussian-like distribution. Experiments are conducted in various task sets.

**Strengths:**

1. The natural property of Perlin Noise such as smoothness can be particularly helpful for robotics tasks as smoother actions are usually perferred due to safety concern.
2. The authors have an extensive proof of the concepts mentioned in the paper.
3. Experiments are carried out in different tasks.

**Weaknesses:**

1. Clarity is my biggest concern. Many math symbols, terms and performance criterion are undefined. Such as t and i in line 211 or the Mean squared Jerk are not properly defined. The procedure of the whole scheme is shown in Fig. 4. However, how normalize function is calculated exactly and what hyperparameters are there are not explicitly written.
2. The results produced by the authors are also different from existing work. Pink noise [1] shows that they are outperforming white noise in many tasks and the results are evaluated in a statistical way. But in this work, pink noise seems to be significantly worse than white noise. The overall performance of Perlin noise and white noise in terms of rewards is also very close to each other.
3. The authors claim that Perlin noise can be used for hard-exploration tasks. But in the selected tasks, there aren't many tasks involving sparse rewards.
4. Smoothness would be the key benefits of using Perlin Noise in robotics. Thus, to fairly compare the benefits, it would be more interesting to compare white/pink noise with a simple PID as the filter.
5. Computational cost is not mentioned. How much extra compute cost would Perlin noise bring?
6. No ablation study has been carried out to show the robustness of Perlin noise

[1] Eberhard, Onno et al. “Pink Noise Is All You Need: Colored Noise Exploration in Deep Reinforcement Learning.” International Conference on Learning Representations (2023).

**Questions:**

Some questiosn are mentioned in the weakness part.
1. Could you please briefly describe the hyperparameter tuning for Perlin noise and other baselines? This would also help for understanding why Pink noise performs so badly.
2. Could you further elaborate how normalize function is in practice calculated?
3.  How much differences in terms of training wallclock time does perlin and other baselines have?
4. How robust is Perlin noise when different t_offset or t_speed are picked?

---

> ### Author Response · Authors · 2024-11-18
>
> Thank you for your thoughtful feedback and for highlighting areas where additional clarification could improve our work. We’ve provided responses below, and please don’t hesitate to ask if more questions arise.
>
> ### Weaknesses:
> > “Clarity is my biggest concern. Many math symbols, terms and performance criterion are undefined. Such as t and i in line 211 or the Mean squared Jerk are not properly defined.”
>
> `i` is the action dimension index, `t` is the timestep index. These definitions should not have been omitted, we will add them. We thank the reviewer for this important notification. We will also formalize Mean Squared Jerk in the appendix. Jerk is the derivative of acceleration. We provide the mean of the squares of the calculated jerks over a series of rollouts during deterministic evaluation in the environments at regular intervals during training.
>
> > “The results produced by the authors are also different from existing work. Pink noise [X] shows that they are outperforming white noise in many tasks and the results are evaluated in a statistical way. But in this work, pink noise seems to be significantly worse than white noise.”
>
> The referenced work tested Pink noise (Colored Noise with $\beta = 1$) in an off-policy setting, benchmarked using SAC. As pointed out in 3.3, one of the problems with Colored Noise is the inability to actually stay on-policy. We expect this to be less of an issue with algorithms designed for off-policy usage. We refer to [1] as a previous work investigating Colored Noise in an on-policy setting. Here Pink ($\beta = 1$) was also found to perform rather badly, the authors instead recommend Colored Noise with $\beta = \frac{1}{2}$, which we refer to as 'HalfBeta' in our work.
>
> > “The authors claim that Perlin noise can be used for hard-exploration tasks. But in the selected tasks, there aren't many tasks involving sparse rewards.”
>
> We focused on environments with task-level exploration, such as those provided by MujocoMaze and our custom harder mazes. Some of the tested MujocoMaze environments use sparse rewards, we refer to the MujocoMaze documentation regarding the exact reward definitions: https://github.com/kngwyu/mujoco-maze. We generally found simply converting existing benchmark environments into sparse-reward environments to be ill-suited to test exploration performance as these were build with current algoritms in mind, leading to solvability issues across the board when just sparsifying the rewards. We further believe our experimental evaluation to be quite broad already.
>
> > “Smoothness would be the key benefits of using Perlin Noise in robotics. Thus, to fairly compare the benefits, it would be more interesting to compare white/pink noise with a simple PID as the filter.”
>
> While it is true that a PID-based approach can sometimes be applied, as shown in the "Walking in the Park" paper [4], where a policy network predicts motor positions and a PID controller generates the actions to achieve them, this setup relies on access to specific internal states of the environment and their response to actions. In RL, such assumptions are not part of the problem definition, making PID control incompatible with standard RL environments.
>
> Our work aims to remain fully compatible with RL environments, which provide only observations and rewards, without requiring modifications or access to internal environment states. Our method is designed as a drop-in replacement for exploration strategies, ensuring broad applicability without additional demands on the environment.
>
> We would be happy to discuss specific test cases where you believe a PID-based comparison could provide valuable insights.
>
> > “Computational cost is not mentioned. How much extra compute cost would Perlin noise bring?”
>
> From a formal perspective, the Complexity is O(n) in the number of action dimensions. (Or more precisely $O(a \cdot n \cdot 2^n)$, where a is the action dimensionality and n is the dimensionality of the spanned Perlin noise, which is always 2 in our case.) From a practical perspective, the provided reference implementation is using a perlin implementation written in vanilla python (no numpy). The training time required per generation is still almost identical between perlin and the other noises. It would be possible to increase the performance significantly by using an efficient implementation.
>
> We thank the reviewer for this important feedback, we have extended the document with a description of the complexity of our sampling mechanism (Section 3.1, line 230)
>
> > “No ablation study has been carried out to show the robustness of Perlin noise.”
>
> We kindly refer the reviewer to the Questions section of our response as we believe that their question was answered there.
>
> [Continued in second post due to character limit]

---

> > ### Author Response · Authors · 2024-11-18
> >
> > [This post continues our response from the last post]
> >
> > ### Questions:
> > > “Could you please briefly describe the hyperparameter tuning for Perlin noise and other baselines? This would also help for understanding why Pink noise performs so badly.”
> >
> > In Section 4, we describe how the hyperparameters (HPs) were found:
> >
> > We tried to limit per-environment and per-suite hyperparameter tuning to avoid overfitting and ensure a fair comparison across methods. Instead, we based our hyperparameter choices on prior works. For PyBullet, we followed the hyperparameters used by Raffin et al. (2022) [2]. For MetaWorld, we adapted the settings from Li et al. (2024) [3]. For general environments such as MujocoMaze and DMC, we relied on the findings of Hollenstein et al. (2024) [1]. Tables of hyperparameters can be found in Appendix C.
> >
> > > “Could you further elaborate how normalize function is in practice calculated?”
> >
> > We thank the reviewer for this question and in addition to our answer kindly refer to our reference implementation. The described process to derive a normalization function is fairly complex in order to be formally sound. As mentioned in the last sentence of Section 3.2, an expansion to first order was found to be sufficient:
> > The normalization function used in the experiments is just a linear function as given by $Normalize(x) = x \cdot 0.57079633$. As shown in Section 3.3, this turns out to be sufficient in practise for accurate normalization, higher order momenta fall right into place on their own. As we have no proof for the sufficience of the linear model, we provide a general explaination of how this normalization could be derived even if Perlin would have turned out to be more ill-behaved in practise instead.
> >
> > > “How much differences in terms of training wallclock time does perlin and other baselines have?”
> >
> > As mentioned before, there is no significant difference. Our wallclock time is dominated by the time per step of the environment simulation. The math behind Perlin is trivial and can be implemented very cheaply.
> >
> > > “How robust is Perlin noise when different t_offset or t_speed are picked?”
> >
> > $t_{offset}$ is only required since we sample all action dimensions from a single shared 2D Perlin. Different implementation choices would allow removing it completely. It must be large enough to ensure the action disturbances are not correlated. We further prefer our sampling lines not falling to close to having an exact integer coordinate, as Perlin has some artifacts at these locations. (Known issue when using Perlin for Computer Graphics). We therefore do not regard $t_{offset}$ as a HP, but rather as an implementation detail. $t_{offset} = \pi$.
> >
> > We will work on an ablation study of $t_{speed}$ and will update the paper once we get results during the rebuttal period.
> >
> > Thank you once again for helping us enhance the clarity and rigor of our paper. We hope these responses provide the clarification you were looking for.
> >
> > [1]: Jakob Hollenstein, Georg Martius, and Justus Piater. Colored Noise in PPO: Improved Exploration and Performance through Correlated Action Sampling. Proceedings of the AAAI Conference on Artificial Intelligence, March 2024. arXiv:2312.11091.
> >
> > [2]: Antonin Raffin, Jens Kober, and Freek Stulp. Smooth exploration for robotic reinforcement learning. In Conference on robot learning, pp. 1634–1644. PMLR, 2022.
> >
> > [3]: Ge Li, Hongyi Zhou, Dominik Roth, Serge Thilges, Fabian Otto, Rudolf Lioutikov, and Gerhard Neumann. Open the Black Box: Step-based Policy Updates for Temporally-Correlated Episodic Reinforcement Learning. In The Twelfth International Conference on Learning Representations, ICLR 2024, Vienna, Austria, May 7-11, 2024. arXiv:2401.11437.
> >
> > [4]: Smith, Laura & Kostrikov, Ilya & Levine, Sergey. (2022). A Walk in the Park: Learning to Walk in 20 Minutes With Model-Free Reinforcement Learning. 10.48550/arXiv.2208.07860.

---

> ### Comment · Reviewer_nqa9 · 2024-11-24
>
> I first would like to thank the authors' efforts on providing more information, which helped me a lot for understanding the paper. I thus, have increased my score. But I do believe that this work still needs more polish, which could not be simply addressed during the rebuttal phase as they might change the scope of the paper. Here are some of my suggestions:
>
> - Currently, the paper title is too ambitous given the results we have seen. Another reviewer has also pointed out that as a general exploration algorithm, applying to both off-policy and on-policy algorithms would be necessary.
> - In the appendix, in some of the environments, white noise could still outperform Perlin noise. A proper explaination of these failure cases would also be great for the practationer.
> - The paper writing should be improved.
>     - Visualization: Legends/x,y-axis label/ticks font size are too small. Unclear definition of the figure, such as the blue dot in fig. 1.
>     - The normalization function and its calculation are unclear to me when reading first through it.
> - For on-policy algorithms, such as PPO, the number of parallel enviroments would also be one essential parameter to investigate. Especially, in the current version, only on-policy is tested and all benchmarks are mostly robotics tasks which actually require smoothness. Then it is straightforward to me to see how perlin noise performs when number of envs in parallel increases since in robotics, parallel training is a common approach.

---

### Official Review · Reviewer_X2qD · 2024-10-30

**Soundness:** 3
**Presentation:** 3
**Contribution:** 2
**Rating:** 6
**Confidence:** 3

**Summary:**

This paper introduces Perlin noise to enhance stochastic policy distributions in reinforcement learning (RL) by encouraging smoother exploration. Perlin noise assigns random gradient vectors to points on a grid and interpolates between them, creating continuous transitions across space, a technique commonly used in computer graphics. The paper details the process of sampling Perlin noise and how it can be leveraged to improve RL policies.

**Strengths:**

The paper leverages the Perlin noise, which is commonly used in computer graphics to generate smooth textures, effectively generate smooth and temporally related noise for RL policies. The Perlin is introduced mainly aiming to generate smoother and more coherent trajectories, thus leading to more nature movements, which is very important in robotics.

**Weaknesses:**

1. The paper claims that using Perlin noise broadens exploration, but this aspect is neither theoretically justified nor sufficiently demonstrated experimentally.

2. While the paper compares several noise types, all approaches focus on adding noise to stochastic policy distributions. Many other exploration methods exist, such as entropy-regularized exploration [1], curiosity-driven exploration [2], intrinsic motivation-based approach [3], reward shaping [4], and novelty-guided exploration [5]. I believe comparisons with some other advanced exploration strategies besides the policy noise based approaches are necessary to fully demonstrate the advantages of Perlin noise in RL.

[1] Haarnoja, T., Zhou, A., Abbeel, P., and Levine, S. Soft actor critic: Off-policy maximum entropy deep reinforcement learning with a stochastic actor. In International Conference on Machine Learning, pp. 1861–1870. PMLR, 2018.

[2] Pathak, Deepak, et al. "Curiosity-driven exploration by self-supervised prediction." International conference on machine learning. PMLR, 2017.

[3] Burda, Y., Edwards, H., Storkey, A., and Klimov, O. Exploration by random network distillation. In International Conference on Learning Representations, 2018.

[4] Devidze, R., Kamalaruban, P., and Singla, A. Exploration guided reward shaping for reinforcement learning under
 sparse rewards. Advances in Neural Information Processing Systems, 2022.

[5] Tang, H., Houthooft, R., Foote, D., Stooke, A., Xi Chen, O., Duan, Y., Schulman, J., DeTurck, F., and Abbeel, P. #
 exploration: A study of count-based exploration for deep reinforcement learning. Advances in Neural Information Processing Systems, 2017.

**Questions:**

1. Regarding Figure 5, I understand that the sampled action trajectories are drawn from a fixed underlying distribution, and it is evident that Perlin noise generates much smoother trajectories. However, these smooth trajectories (e.g., over 100 actions) are based on a fixed underlying distribution. In real RL learning or inference, the stochastic policy distribution $\pi(\cdot | s_t)$ is conditioned on the current state, meaning the policy distribution changes at each step. In this scenario, does the sequence smoothness still hold? In other words, would generating Perlin noise for a changing policy trajectory still result in smooth action sequences?

2. Does the use of smooth action trajectories imply that action changes are relatively slow? Could this affect the speed of learning or inference? For instance, in scenarios that require rapid action adjustments (such as emergency braking in autonomous driving), would this smoothness impact the effectiveness of the policy?

3. Besides generating smooth action sequences, does Perlin noise effectively increase the diversity of the sampled actions? In model-free RL, efficient exploration, i.e., collecting more diverse samples, is critical for improving sample efficiency. How does the introduction of Perlin noise influence the breadth of exploration?

I would like to increase the scores if these concerns are addressed.

---

> ### Author Response · Authors · 2024-11-18
>
> Thank you for your valuable feedback and for taking the time to engage so thoughtfully with our work. We’ve provided responses to your concerns below and hope these address the questions you raised. Please feel free to reach out if additional clarification is needed.
>
> ### Questions:
> > “Regarding Figure 5, I understand that the sampled action trajectories are drawn from a fixed underlying distribution, and it is evident that Perlin noise generates much smoother trajectories. However, these smooth trajectories (e.g., over 100 actions) are based on a fixed underlying distribution. In real RL learning or inference, the stochastic policy distribution is conditioned on the current state, meaning the policy distribution changes at each step. In this scenario, does the sequence smoothness still hold? In other words, would generating Perlin noise for a changing policy trajectory still result in smooth action sequences?”
>
> We refer to our 'Response to Reviewer Feedback' for an explanation of what smoothness we are capable of providing. Our Perlin-based disturbances have no impact on the policy mean generated by the policy NN. PPO (according to the original paper and reference implementation) does not contextually parameterize the variance; it is modeled as a simple parameter. Therefore, the variance tends to slowly decrease across the training generations. We guarantee that the applied disturbances are smooth. Whether this also results in actually smooth actions is further dependent on the policy parameters. If the NN learns to parameterize high-frequency actions, the resulting trajectories will also be jerky. We provide measured mean squared jerk of the actions as a measure of action smoothness in the appendix.
>
> We generally observe that the smoothness in action disturbances also leads to higher smoothness of sampled actions during training as can be seen in Appendix B.
>
> > “Does the use of smooth action trajectories imply that action changes are relatively slow? Could this affect the speed of learning or inference? For instance, in scenarios that require rapid action adjustments (such as emergency braking in autonomous driving), would this smoothness impact the effectiveness of the policy?”
>
> Our idea was in parts inspired by recent progress with movement primitive-based trajectory-centric RL. These techniques often achieve higher performance, but at the cost of being unable to generate high-frequency actions (and not being directly applicable to normal step-based RL envs). Remaining in the step-based paradigm, but carrying over the smooth trajectory disturbance behavior from these (as we do with Perlin-based exploration), we see that we can increase the performance of such traditional step-based algorithms as PPO. We also carry over a bias for smooth actions. However, contrary to MP RL, we are not unable to parameterize high-frequency actions; we merely disfavor them during training. This will still harm the ability (extend required time) to converge for certain tasks that require such high-frequency actions (we see this as the reason for the sub-par performance indicated in Figure 12(e): DMC Hopper Hop), but contrary to MP RL, we do remain able to learn high-frequency motions if required eventually. Exploration methods tend to be disabled when deploying policies (switching to deterministic policy evaluation), so we have no impact on the performance of a deployed system, apart from maybe having helped the agent to learn a better policy.
>
> > “Besides generating smooth action sequences, does Perlin noise effectively increase the diversity of the sampled actions? In model-free RL, efficient exploration, i.e., collecting more diverse samples, is critical for improving sample efficiency. How does the introduction of Perlin noise influence the breadth of exploration?”
>
> We believe it is highly depending on the definition of 'diversity'. As can be seen in Figure 1, we achieve a higher coverage of the state space. We would regard this as an indication of higher diversity of trajectories. But this requires looking at the diversity of whole trajectories. For a per-action treatment, one could also try to formalize the 'diversity' of an action via its entropy. In this case, Perlin-based exploration will actually decrease the 'diversity', as we favor consistent actions to follow one another, therefore decreasing the entropy for each individual step. Perlin restricts the diversity of any individual action in favor of higher diversity of complete interactions with the environment by being able to escape from local optima and expanding the scope of states reached.
>
> [Continued in second post due to character limit]

---

> > ### Author Response · Authors · 2024-11-18
> >
> > [This post continues our response from the last post]
> >
> > ### Weaknesses:
> > > “The paper claims that using Perlin noise broadens exploration, but this aspect is neither theoretically justified nor sufficiently demonstrated experimentally.”
> >
> > We agree with the reviewer that we haven't provided any theoretical proofs or justifications. If the reviewer is aware of any theoretical analysis tools we are happy to investigate them. Yet, our analysis show how the more consistent smooth disturbances generated using Perlin-based disturbances yield higher state-space reach (e.g. Figure 1). While we see the benefit in a further formal treatment of this line of reasoning, we believe that commonly used formalizations (like action-wise entropy) to be unable to capture actual explorative capabilities. We instead validate the claim of better exploration in our broad experiments.
> >
> > > “While the paper compares several noise types, all approaches focus on adding noise to stochastic policy distributions. Many other exploration methods exist, such as entropy-regularized exploration, curiosity-driven exploration, intrinsic motivation-based approach, reward shaping, and novelty-guided exploration.”
> >
> > We constrained ourselves to Maximum Entropy (entropy-regularized exploration) as method for modulating exploration (but found best overall performance with an entropy coefficient of 0, as is documenetd in the HP tables in the Appendix). We excluded intrinsic motivation and similar techniques from our experiments since we believe exploration modulation to be a separate issue, not directly related to the question about how trajectories are ensured to be explorative, which was the focus of our work. Techniques like novelty-guided exploration are not applicable to continuous state / observation spaces without leaving the regime of model-free RL, and were therefore also not explored as direct comparisons.
> >
> > > “Concerns regarding whether smoothness of action sequences holds when the policy distribution changes at each step.”
> >
> > This question is very related to the questions in the Questions section. We kindly refer the reviewer to our answers there.
> >
> > > “Questions whether smoother trajectories might slow down learning or inference, particularly in scenarios requiring rapid response.”
> >
> > This question is very related to the questions in the Questions section. We kindly refer the reviewer to our answers there.
> >
> > We hope these responses help clarify the points you raised and enhance our work’s transparency. Don’t hesitate to reach out with any additional questions.

---

> > > ### Comment · Area_Chair_hQGC · 2024-11-24
> > > **Please respond to rebuttal ASAP**
> > >
> > > Dear reviewer,
> > > The process only works if we engage in discussion. Can you please respond to the rebuttal provided by the authors ASAP?

---

> > > ### Comment · Reviewer_X2qD · 2024-11-25
> > >
> > > Many thanks for the response. Many of my concerns are explained, and I'd like to increase the score to 6.
> > >
> > > However, there are some improvements that can be made and I highly expect these can be incorporated into the paper:
> > >
> > > 1. As the paper mainly claims two points: one is Perlin noise can smooth the actions, which has been well demonstrated and explained, while another one is Perlin noise can broaden the exploration, for this point, we can only see in Figure 1, therefore, more related experiments can be added. Because this is an "exploration" based study, comparison with a baseline that encourages exploration is inevitable. Actually, there are many novelty-based exploration methods that can work in continuous state and action spaces, within the model-free RL, like the representative RND, (my suggested [3][4][5] references all can work on that). If one or two non-noise-based exploration approaches can be compared, the work will be much more convincing.
> > >
> > > 2. As this is an "exploration" study, the evaluations can be conducted in some sparse-reward environments to further show its effectiveness (I may have overlooked whether or not the experimental environment rewards are sparse, so if I'm wrong please ignore this point).

---

### Official Review · Reviewer_qdjK · 2024-11-03

**Soundness:** 3
**Presentation:** 4
**Contribution:** 1
**Rating:** 3
**Confidence:** 5

**Summary:**

This paper proposes using Perlin noise as a drop-in replacement to other forms of noise for any RL algorithm, and demonstrates improvements in exploration in terms of both final performance and smoothness on various tasks in many domains.

**Strengths:**

- The paper is well written with extensive evaluations on various tasks and in various domains.
-  Naively sampling perlin noise and applying it to the actions of the policy makes it impossible to use the reparameterization trick to compute gradients as in SAC. The authors propose a novel approach of first applying perlin noise to the actions and then modeling the resulting policy using a gaussian distribution using the “Normalize” trick. I agree with the authors that this approach is novel and makes it fairly straightforward for any RL practitioner to use it as a drop-in replacement for other forms of exploration noise.
- The authors demonstrate good performance improvements on select difficult exploration tasks such as Ant maze (Fig. 7) and provide empirical evidence that Perlin noise leads to smoother action trajectories (Fig. 6).

**Weaknesses:**

- Perhaps my main concern with this paper is that, at a high level, it simply proposes and investigates a different noise function for PPO. While simple solutions especially if they solve major problems are not necessarily bad, the novelty and technical contribution here is quite limited.

- Since the claim is that Perlin noise provides more smooth, structured exploration independent of the algorithm or task, it seems necessary to have at least one more algorithm beyond PPO to validate this claim.

-  Figure 5 needs some context on what the task is, what kind of policy was used to collect the trajectory (is it a random policy? Trained policy? Hand-engineered one?), etc. For example, PPO has a learnable action standard deviation parameter that generally tends towards 0 i.e. a deterministic policy as the policy converges. I would then expect that the trajectory, regardless of what kind of noise is applied, to look relatively smooth. It is impossible to make any conclusions without further context here.

- Final minor complaint: On lines 59-60 the authors contrast against intrinsic-motivation and trajectory-level noise methods by saying  *“However, these methods require additional treatment of the underlying optimization method by changing the objectives or introducing higher-level abstractions of the actions.”*.  Why is this a problem? This is similar to saying “PPO does better than REINFORCE, however it requires additional treatment of the underlying optimization method”. If that “underlying treatment” results in a fundamentally better algorithm, then the research community will adopt this new approach. This, in my opinion, is not a bad thing but rather the natural progression of a field.

Overall, I appreciate the simplicity of the method, the extensive experiments on a variety of domains, and generally agree with the conclusions the authors draw. However, I don't believe that proposing a different noise function for just PPO is sufficient for publication as a conference paper. Accepting this paper sets a precedent for others to just try different noise functions on different algorithms on different tasks, and there are a countless number of ways to achieve "novel" results this way, none of which actually help the field progress. Perhaps a comprehensive study of many different noise functions on different algorithms and tasks, and a deeper analysis of why certain types of noise do well on certain algorithms / tasks and not others would warrant a full paper. The work the authors present here would do better in the blogpost or workshop track.

**Questions:**

No questions at this time.

---

> ### Author Response · Authors · 2024-11-18
>
> Thank you for your careful reading of our work and for your insightful feedback. We appreciate the opportunity to clarify and expand upon our contributions. Should further questions arise, we would be more than happy to provide additional clarification.
>
> ### Weaknesses:
> > “Perhaps my main concern with this paper is that, at a high level, it simply proposes and investigates a different noise function for PPO. While simple solutions especially if they solve major problems are not necessarily bad, the novelty and technical contribution here is quite limited.”
>
> Yes, it’s simple—like PPO’s advantage over TRPO, simplicity here is intentional and valuable. Perlin noise adds novel structured, smooth exploration that current noise types lack, ensuring fluid motions and overcoming local optima with minimal changes. While many complex methods can boost performance, they often aren’t adopted due to implementation and tuning difficulty. In contrast, we see our approaches simplicity and drop-in usability as an advantage, making it feasible for practical use and further research.
>
> > “Since the claim is that Perlin noise provides more smooth, structured exploration independent of the algorithm or task, it seems necessary to have at least one more algorithm beyond PPO to validate this claim.”
>
> We agree with the reviewer that further research across algorithms is valuable. In this work, we focused on on-policy methods, as they tend to suffer more from converging on local optima and imitating jerky actions seen during training. PPO was chosen as the de facto standard for on-policy, but future work could explore Perlin noise’s impact on other on- and off-policy methods. But we see this as outside the scope of this paper and this rebuttal phase.
>
> > “Figure 5 needs some context on what the task is, what kind of policy was used to collect the trajectory (is it a random policy? Trained policy? Hand-engineered one?), etc.”
>
> Other reviewers asked similar questions. We refer to our 'Response to Reviewer Feedback' for an answer and have updated the paper for more clarity (changes also described in the 'Response to Reviewer Feedback').
>
> > “On lines 59-60 the authors contrast against intrinsic-motivation and trajectory-level noise methods by saying ‘However, these methods require additional treatment of the underlying optimization method by changing the objectives or introducing higher-level abstractions of the actions.’ Why is this a problem?”
>
> We regard not requiring more extensive adjustments as a benefit, as it allows using our technique as a drop-in replacement. Often, such adjustments also result in new additional problems, as the inability to generate high-frequency actions we observe with trajectory-level RL.
>
> Thank you again for your constructive feedback and for helping us refine our paper. We hope these responses clarify our approach and motivation.

---

> > ### Comment · Area_Chair_hQGC · 2024-11-24
> > **Please respond to rebuttal ASAP**
> >
> > Dear reviewer,
> > The process only works if we engage in discussion. Can you please respond to the rebuttal provided by the authors ASAP?

---

> > ### Comment · Reviewer_qdjK · 2024-11-27
> > **Response to Authors**
> >
> > Thank you for taking the time to respond to my questions and concerns. I will respond to comments individually.
> >
> > **Yes, it’s simple—like PPO’s advantage over TRPO, simplicity here is intentional and valuable. Perlin noise adds novel structured, smooth exploration that current noise types lack, ensuring fluid motions and overcoming local optima with minimal changes**
> >
> > I agree that simpler ideas that are easier to implement, especially those that yield large changes in how the method behaves and performs are particularly valuable. I am not against the idea of proposing simpler ideas if they add a great deal of novelty, insight, and / or performance changes. However, I am not convinced by the authors' response that the proposed change falls into this category. Changing the noise scheme in PPO is not the result of some deep insight but rather a fairly obvious thing to try. As for the performance and behavior of the method, while it does increase the performance slightly on some tasks, it also decreases or produces no change on others, and so the proposed change resembles more of a tradeoff than a paradigm shifting idea that significantly changes the performance and behavior of the algorithm.
> >
> > **PPO was chosen as the de facto standard for on-policy, but future work could explore Perlin noise’s impact on other on- and off-policy methods. But we see this as outside the scope of this paper and this rebuttal phase.**
> >
> > I disagree with this statement. The paper's structure, from the title to the text itself, suggests that Perlin noise is a suitable alternative and "drop-in replacement" for existing RL algorithms in general (ex. lines 69-73). As a reader, I would then expect to see how the proposed change performs on several different RL algorithms. If the authors believe this is out of the scope of this paper, then the title and contents should be changed to reflect this.
> >
> > Overall, I maintain my belief that work like this would be better suited as a workshop paper or blogpost track, with perhaps an investigation into how different noise functions affect different RL algorithms with some deeper analysis warranting a full paper.

---

### Author Response · Authors · 2024-11-18
**Response to Reviewer Feedback and Paper Changelog**

We extend our sincere thanks to all reviewers for their detailed and constructive feedback. Your insights, particularly regarding the need for added context around certain figures and explanations, have highlighted areas where our work can be made clearer and more accessible. We hope that our response below addresses the collective concerns raised, and we look forward to further improving our paper based on these helpful observations.

Since multiple reviewers asked questions regarding Figure 5:
Figure 5 is referenced in Section 3.3, which does contain, what we believed to be a sufficient description. But since multiple reviewers pointed out a lack of context surrounding it, it seems like either the description is not clear or should be repeated in the subtitle of the Figure. A more elaborate explanation:

We can take two separate views of how we use Gaussian action sampling in RL:
 1. We define the policy as a Gaussian Policy. In Vanilla PPO, we span the mean via a NN, the variance is a parametric vector, which is also optimized via gradient descent, but is not dependent on the observation.
 2. We assume the policy NN to provide us with an deterministic action, which we then disturb via a random sample from Gaussian Noise with zero mean.

Mathematically, both views are equivalent; they differ only in where we draw the line of what is part of the policy definition. While the first view is common in the literature, we will adopt the second in our explanation.

In Section 3.3, we investigate the behavior of the tested noises initially in isolation from an RL setup using a static Gaussian Policy (N(0,1)). In an actual RL setup (as evaluated in section 4), this is the noise sample used to disturb the policy mean / deterministic action.
Switching from Colored noises to Perlin noise allows us to ensure, that these disturbances are smooth, slowly varying over time. This is independent of the mean of the action policy: If the NN does not provide smoothly varying means, the resulting action will also always be unsmooth. But following the proposed NN weight initialization of the original PPO implementation, these will be smooth in the beginning and we do not expect the network to learn to parameterize jerky means unless it is required to solve the task.

We worked this and other changes into the document:
- **Added Definitions for \(i\) and \(t\) in Section 3.1 (Line 210):** We have now explicitly defined the indices \(i\) and \(t\) to avoid confusion, ensuring that the notation is clear and consistent throughout the text.

- **Described the Computational Complexity of Perlin Noise Using Our Sampling Mechanism (Section 3.1, Line 230):** We have added an explanation of the computational complexity of our Perlin noise sampling mechanism to Section 3.1. This ensures a clear understanding of the trade-offs and efficiency of the method in terms of computational cost.

- **Expanded Figure 5 and Section 3.3 for Better Clarity:** In response to feedback, we have added additional context and details to Figure 5 and Section 3.3. This aims to make the figure’s purpose and content more transparent.

- **Added Formalization and Calculation Description of 'Mean Squared Jerk' to Appendix (B.1):** To address concerns about clarity in the jerk calculation, we have formally defined the Mean Squared Jerk (MSJ) metric in Appendix B.1.

---

### Meta-Review · Area_Chair_hQGC · 2024-12-20

**Metareview:**

This works proposes to replace standard Gaussian exploration in on-policy RL methods usign Perlin noise based exploration, which is smooth and temporally correlated. The key innovation is to alter the noise structure and then propose a reparameterization with a new "normalize" function that enables the use of a standard likelihood or reparameterization based RL method. The methodology is investigated across several tasks in simulation.

Strengths:
The idea of Perlin noise is neat, seems really useful for most RL problems with temporally extended decision making.
The methodology to enable likelihood/reparameterization based RL is neat!

Weaknesses:
Really the major weakness of the paper is that the empirical results are not strong enough - the gap between Perlin and other methods is relatively small and not very marked.
I think the authors' argument of discounting all intrinsic exploration methods is not a completely justified one. If the argument is that these are complementary methods, this should really be demonstrated in the paper.

Overall the paper holds a lot of merit, but a stronger empirical comparison, more baselines that do exploration and a comparison on sparse reward tasks would add a lot of value.

**Additional Comments On Reviewer Discussion:**

Reviewers brought up a variety of concerns. Reviewer qdjK brought up a philosophical point of whether this is a major enough contribution for a conference because it is a simple change. I don't believe that is an appropriate assessment here, the work needs to add a clear insight and a clear contribution and I believe that the work largely does those besides through the strength of the empirical results. The other reviewers brought up comparisons to intrinsic motivation methods which should really be made in the paper, as well as differences in the numbers as opposed to papers like the pink noise paper. The clarification of on vs off-policy should be made clearly here, and if on-policy is preferred and used, the argument for why should be made here as well.

---

### Decision · Program_Chairs · 2025-01-22

Reject